



# The impact of meltwater discharge from the Greenland ice sheet on the Atlantic nutrient supply to the northwest European shelf

**Moritz Mathis**[1,2] **and Uwe Mikolajewicz**[1]

[1]Ocean Department, Max-Planck-Institute for Meteorology, Bundesstr. 53, 20146 Hamburg, Germany
[2]Helmholtz-Zentrum Geesthacht, Institute of Coastal Research, Max-Planck-Str. 1, 21502 Geesthacht, Germany

**Correspondence:** Moritz Mathis (moritz.mathis@hzg.de)

**Abstract.** Projected future shoaling of the wintertime mixed layer in the northeast (NE) Atlantic has been shown to induce a regime shift in the main nutrient supply pathway from the Atlantic to the northwest European shelf (NWES) near the end of the 21st century. While reduced winter convection leads to a substantial decrease in the vertical nutrient supply and biological productivity in the open ocean, vertical mixing processes at the shelf break maintain a connection to the subpycnocline nutrient pool and thus productivity on the shelf. Here, we investigate how meltwater discharge from the Greenland ice sheet (GIS), not yet taken into account, impacts the mixed layer shoaling and the regime shift in terms of spatial distribution and temporal variability. To this end, we have downscaled sensitivity experiments by a global Earth system model for various GIS melting rates with a regionally coupled ocean–atmosphere climate system model. The model results indicate that increasing GIS meltwater discharge leads to a general intensification of the regime shift. Atlantic subpycnocline water masses mixed up at the shelf break become richer in nutrients and thus further limit the projected nutrient decline on the shelf. Moreover, the stronger vertical nutrient gradient through the pycnocline results in an enhanced interannual variability of on-shelf nutrient fluxes which, however, do not significantly increase variations in nutrient concentrations and primary production on the shelf. Due to the impact of the GIS meltwater discharge on the NE Atlantic mixed layer depth, the regime shift becomes initiated earlier in the century. The effect on the onset timing, though, is found to be strongly damped by the weakening of the Atlantic meridional overturning circulation. A GIS melting rate that is even 10 times higher than expected for emission scenario Representative Concentration Pathway (RCP) 8.5 would not lead to an onset of the regime shift until the 2070s.

## 1 Introduction

Previous climate change impact studies identified a general weakening of the biological productivity of the outer northwest European shelf (NWES) as a regional response to a globally warming climate (e.g., Holt et al., 2012, 2016; Gröger et al., 2013; Wakelin et al., 2015; Schrum et al., 2016; Mathis et al., 2018). The main driver has been attributed to a reduction in the oceanic nutrient import from the adjacent northeast (NE) Atlantic. Under present-day conditions, up to 80 %–90 % of the nutrient inventory on the shelf is advected from the open ocean via cross-shelf break transport of nutrient-rich Atlantic water masses (e.g., Vermaat et al., 2008; Liu et al., 2010; Thomas et al., 2010). The nutrient concentrations of the water masses flushing the shelf are primarily controlled by the maximum depth of the wintertime mixed layer (MLD) west of the shelf break (e.g., Holliday, 2003; Williams et al., 2011; Holt et al., 2012; Gröger et al., 2013; Mathis et al., 2018). The warming and freshening of the upper North Atlantic projected by global climate models induces a weakening of the buoyancy-driven convection and thus a reduction of the wintertime MLD and upper-ocean nutrient concentration (e.g., Yool et al., 2015; Fu et al., 2016; Alexander et al., 2018). Accordingly, previous downscaling simulations mainly based on the Special Report on Emissions Scenarios (SRES) emission scenario A1B indicated that a nutrient decline in the upper NE Atlantic would lead to a similar nutrient decline on the NWES, limiting net primary

production and atmospheric $CO_2$ uptake in open shelf areas (e.g., Holt et al., 2012; Gröger et al., 2013; Wakelin et al., 2015; Schrum et al., 2016). The parent global models used to provide the forcing data, however, did not include interactive ice sheet models and hence miss an important factor in their projections of future changes in the North Atlantic circulation and water mass characteristics: the melting of the Greenland ice sheet (GIS). In particular, the effect of increasing meltwater discharge to the subpolar North Atlantic on the ocean mixed layer shoaling is not taken into account and therefore is not included in the downscaling simulations for the NWES.

Observational data have provided evidence for a significant increase in GIS mass loss during the recent decades (Luthcke et al., 2006; Vaughan et al., 2013; van den Broeke et al., 2017; Chen, 2019). Modeling studies considered by the Fifth Assessment Report of the United Nations Intergovernmental Panel on Climate Change (IPCC AR5) agree that the GIS will continue to decrease in area and volume in a warmer climate as a consequence of increased melting rates not compensated by increased snowfall and amplified by positive feedbacks (Mikolajewicz et al., 2007; Collins et al., 2013; Church et al., 2013; Vizcaino et al., 2015). Enhanced GIS meltwater discharge has been shown to decrease wintertime MLDs in the North Atlantic due to the haline-induced reduction of surface buoyancy (Böning et al., 2016; Oliver et al., 2018). To that effect, numerous other studies suggest that substantial meltwater input to the North Atlantic will weaken the Atlantic meridional overturning circulation (AMOC) or even lead to a collapse (Schiller et al., 1997; Stouffer et al., 2006; Weijer et al., 2012; Sévellec et al., 2017; Liu et al., 2018). The resulting decrease in northward heat transport might further affect the surface buoyancy and thus the permanent stratification and MLD (Hu et al., 2009; Woollings et al., 2012; Wouters et al., 2012; Liu et al., 2017).

In a recent study by Mathis et al. (2019), it has been shown that for a critical ML shoaling according to emission scenario Representative Concentration Pathway (RCP) 8.5, upper North Atlantic water masses lose their dominant influence on shaping biogeochemical conditions on the NWES. This is due to a regime shift in the main supply pathway of Atlantic nutrients to the shelf. In the open ocean, the ML shoaling leads to a reduction of the vertical nutrient supply from intermediate depths to the euphotic zone. At the shelf break, however, various mixing processes maintain a connection to the Atlantic subpycnocline nutrient pool, e.g., due to the interaction of tidal currents with the abrupt change in the topography, internal waves and instabilities of the slope current. Nutrient-enriched water masses mixed up near the shelf break spread to the NWES and lead to a weaker nutrient decline in open shelf areas, compared to the adjacent upper NE Atlantic. As a consequence, the projected weakening of the biological productivity on the shelf is not as strong as would have resulted from a sole influence of upper NE Atlantic water masses. Finally, nutrient concentrations in the upper NE Atlantic reach lower levels than on the shelf, leading to a reversal of the ocean–shelf nutrient gradient and to the development of a nutrient front along the continental margin.

Furthermore, during the shallow-ML regime, the nutrient transport to the NWES is subject to a rapid increase in interannual to multidecadal variability. As near the end of the 21st century the ML in the NE Atlantic becomes as shallow as the shelf edge, upper-ocean conditions are more sensitive to variations in the atmospheric forcing. In particular, a positive MLD anomaly (deepening) leads to an erosion of warm and saline subpycnocline water masses and initiates a positive feedback on the surface heat flux, the upper-ocean buoyancy and the MLD. The resulting enhanced variability of the on-shelf nutrient transport affects the variability in pre-bloom nutrient concentrations and annual primary production on the NWES.

Similar to previous downscaling studies, though, the freshwater cycle in the underlying simulations by Mathis et al. (2019) did not account for a future increase in GIS meltwater discharge, ignoring its impact on the density distribution in the upper North Atlantic. Hence, it is not yet clear how and to what extent GIS meltwater discharge impacts the proposed regime shift in the Atlantic nutrient supply to the NWES. In particular, it has been shown by Mathis et al. (2019) that during the shallow-ML regime the transfer of anomalies in the upper NE Atlantic circulation and water mass characteristics to the NWES becomes undermined by the associated shift in the dominant ocean–shelf exchange processes from horizontal advection to vertical mixing. In the present study, our general goal is to explore the influence of GIS meltwater discharge on the signal transfer to the NWES and thus to advance our understanding of the coupling mechanisms in the transition zone between the open ocean and the shallow shelf under a strong radiative forcing scenario. Specifically, we focus on the following questions: Does the regime shift become initiated earlier in the 21st century due to the impact on the MLD? Does the ML become shallower than the depth of the shelf edge, providing a direct connection between the shelf and the Atlantic subpycnocline nutrient pool? Does the enhanced variability of the nutrient concentrations at the shelf break become even stronger due to the impact on the permanent stratification? If such changes are found, what are the consequences for the nutrient budget and primary production in open shelf areas such as the northern North Sea, which are predominantly influenced by Atlantic inflow? In the present study, we address these questions by an investigation of the impact of GIS meltwater discharge on both the mean Atlantic nutrient supply to the NWES and its temporal variability on interannual to multidecadal scales.

The original study by Mathis et al. (2019), henceforth referred to as M19, was based on a dynamical downscaling of global climate projections for emission scenario RCP8.5 by the Max-Planck-Institute Earth System Model (MPI-ESM-LR). It was shown that the global model was not able to capture the drastic changes in the on-shelf nutrient transport

near the end of the 21st century. By contrast, pre-bloom nutrient concentrations on the NWES simply reflected the continuous upper-ocean nutrient decline of the NE Atlantic due to an underrepresentation of local processes relevant to adequately simulating a comparatively small region such as the NWES. Therefore, in the present study, we utilize the same high-resolution regionally coupled climate system model as in M19 and conducted a set of downscaling simulations for emission scenario RCP8.5 with additional freshwater runoff along the coast of Greenland. To account for uncertainties due to natural variability, we performed ensembles of three realizations for experiments with and without GIS meltwater discharge as well as a pre-industrial control run. The downscaling model system is briefly described in Sect. 2.1, and the experiment design is introduced in Sect. 2.2. Results are presented and discussed in Sects. 3 and 4, respectively. Concluding remarks are given in Sect. 5.

## 2   Methods

### 2.1   Model description

The downscaling simulations presented here are performed with a regionally coupled ocean–atmosphere climate system model, consisting of the global Max-Planck-Institute Ocean Model (MPIOM) with a zoom on the NWES, the Hamburg Ocean Carbon Cycle Model (HAMOCC), the Regional Atmosphere Model (REMO) and the global Hydrological Discharge Model (HD). A schematic of the ocean grid and coupling domain is shown, e.g., in M19. Simulated physical and biogeochemical conditions on the NWES have been evaluated in various studies employing MPIOM/HAMOCC in coupled and uncoupled modes (Gröger et al., 2013; Mathis et al., 2015, 2018, 2019; Pätsch et al., 2017; Hátún et al., 2017).

MPIOM (Maier-Reimer, 1997; Marsland et al., 2003; Jungclaus et al., 2013) is the ocean–sea ice component of the global Max-Planck-Institute Earth System Model (MPI-ESM). The primitive equations of oceanic motion are discretized on an Arakawa C grid with $z$ coordinates and free surface. Hydrostatic and Boussinesq approximations are applied. A second-order total variation diminishing scheme (Sweby, 1984) is used to simulate tracer and momentum advection. Vertical mixing is implemented after Pacanowski and Philander (1981), with an additional parameterization for wind-induced stirring (Marsland et al., 2003). The full lunisolar tidal potential is calculated according to Thomas et al. (2001).

Using a global ocean component in the downscaling model avoids problematic influences of open lateral boundary conditions necessary to run conventional regional ocean models (Mathis et al., 2018). Higher grid resolutions in the subpolar North Atlantic and NWES regions are achieved by using a stretched grid configuration with non-diametrical poles lo-

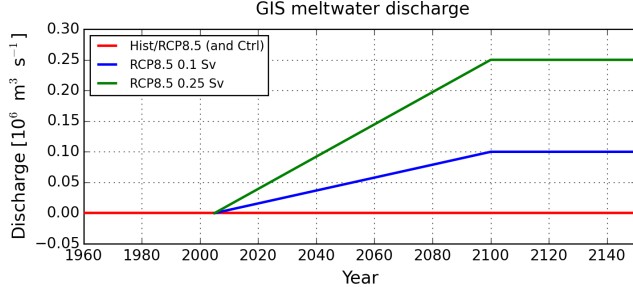

**Figure 1.** Prescribed idealized meltwater discharge from the Greenland ice sheet to the North Atlantic. The distribution along the coast of Greenland follows the climatologies by Bamber et al. (2012) and Martin et al. (2019).

cated over central Europe and North America (Mathis et al., 2018, 2019). In the present study, MPIOM has been run at a nominal horizontal grid resolution of 0.6°, yielding mesh sizes of about 5 to 12 km on the NWES. The vertical is resolved by 30 levels with eight levels in the upper 100 m.

Biogeochemical processes in the ocean are simulated by HAMOCC (Maier-Reimer et al., 2005; Ilyina et al., 2013). Marine biology dynamics is represented by nutrients, phytoplankton, zooplankton, detritus and dissolved organic matter (Six and Maier-Reimer, 1996). Biogeochemical cycles and trophic levels are connected by nutrient uptake and remineralization of organic matter (Six and Maier-Reimer, 1996; Kloster et al., 2006). A sediment module accounts for deposition and dissolution processes as well as the biogeochemistry of pore water in the upper bioturbated layers (Heinze et al., 1999). Sediment resuspension is based on the concept of incipient motion (Wilcock, 2004; Mathis et al., 2019). Tracer transport is simulated by MPIOM.

Over the EURO-CORDEX domain (e.g., Kotlarski et al., 2014), MPIOM is interactively coupled with REMO (Jacob and Podzun, 1997; Jacob et al., 2001) at a coupling time step of 1 h. REMO has been run at a uniform horizontal resolution of about 25 km and 27 hybrid levels. Sea surface fluxes of heat, momentum and freshwater are transferred to MPIOM by the OASIS3 coupler (Valcke, 2013). In turn, sea surface temperature and sea ice cover and thickness are passed from MPIOM to REMO. At the land–atmosphere interface, a simple bucket scheme is applied for the treatment of soil hydrology. Freshwater drainage and runoff are calculated by the HD model (Hagemann and Dümenil-Gates, 2001).

As forcing data for the downscaling model, we use 6-hourly atmospheric output from the global MPI-ESMCEI in its low-resolution version (LR; Giorgetta et al., 2013). The data are used as lateral boundary conditions for REMO and as input for the calculation of atmosphere–ocean fluxes by MPIOM outside the domain covered by REMO.

**Table 1.** Overview of the experimental setup. Each realization has been performed with the global MPI-ESM-LR and dynamically downscaled with the regionally coupled ocean–atmosphere climate system MPIOM/HAMOCC/REMO model CE2.

| Experi-ment | Simulation period | Emission scenario | GIS meltwater discharge | Number of realizations |
|---|---|---|---|---|
| C0 | 1920–2150 | Pre-industrial | 1920–2150 0.0 Sv | 1 |
| E0 | 1920–2150 | Historical/ RCP8.5 | 1920–2150 0.0 Sv | 3 |
| E010 | 1920–2150 | Historical/ RCP8.5 | 1920–2005 0.0 Sv 2006–2100 lin. increase 2101–2150 0.1 Sv | 3 |
| E025 | 1920–2150 | Historical/ RCP8.5 | 1920–2005 0.0 Sv 2006–2100 lin. increase 2101–2150 0.25 Sv | 1 |
| E100 | 1920–2150 | Historical/ RCP8.5 | 1920–2005 0.0 Sv 2006–2100 lin. increase 2101–2150 1.0 Sv | 1 |

## 2.2 Experiment design

The future regime shift in Atlantic nutrient supply to the NWES (M19) has been identified for emission scenario RCP8.5. To investigate the impact of GIS meltwater discharge in the present study, we therefore conduct RCP8.5 simulations with various amounts of additional freshwater runoff along the coast of Greenland (Fig. 1 and Table 1). The spatial distribution of the runoff (Fig. 2) follows the climatology by Bamber et al. (2012), based on satellite observations and regional climate modeling. The seasonal cycle (Fig. A1) has been derived by Martin et al. (2019). The annual mean GIS freshwater flux according to this study corresponds to 0.05 Sv and is comparable to the estimate of about 0.04 Sv starting from the year 2010 by Yang et al. (2016) and Bamber et al. (2018). In our simulations, the prescribed freshwater fluxes enter the surface layer of the ocean model, thus ignoring that many marine-terminating outlet glaciers have a grounding line depth several hundred meters below sea level (An et al., 2017; Morlighem et al., 2017).

All experiments are performed with both the global MPI-ESM-LR model to create forcing data and the MPIOM/HAMOCC/REMO downscaling model. The simulations cover the period 1920–2150 in consistency with M19. The MPI-ESM-LR simulations are initialized by 1920 conditions of the MPI-ESM-LR consortium simulations which contributed to CMIP5 (Coupled Model Intercomparison Project, phase 5). The original CMIP5 simulations could not be used here because they were run on a former high-performance computer (HPC) and hence are inconsistent with our GIS discharge experiments. A bitwise reproduction of a simulation is not possible on different HPCs, leading to independent realizations even when started from identical initial conditions. To minimize the effect of long-term model drift, the MPIOM/HAMOCC/REMO simulations are initial-

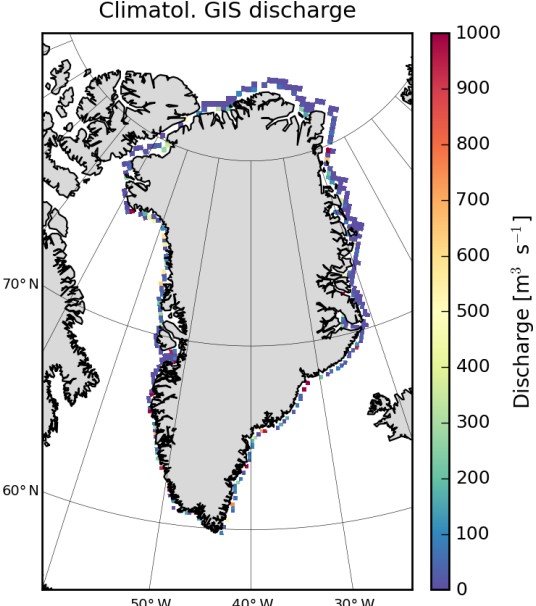

**Figure 2.** Spatial distribution of annual mean GIS discharge following the climatologies by Bamber et al. (2012) and Martin et al. (2019). The pronounced seasonal cycle is shown in Fig. A1.

ized by the end of the pre-industrial control simulation (year 2150) used in M19.

Experiment C0 is the control run at constant pre-industrial atmospheric greenhouse gas concentrations without additional GIS meltwater discharge. This experiment allows us to account for remaining model drift as well as to distinguish anthropogenic climate change signals from natural variability.

Experiment E0 is a combination of the historical period 1920–2005 and the RCP8.5 scenario period 2006–2150. Also, here, no additional GIS meltwater discharge is taken into account, and hence this experiment is comparable to the original simulation by M19. However, in the present study, we have performed an ensemble of three realizations (Table 1) differing in the initial conditions used for the parent MPI-ESM-LR simulations. This allows us to assess the influence of internal variability of the forcing global model.

In experiment E010, a linear increase of GIS meltwater discharge from 0 Sv in the year 2006 to 0.1 Sv in 2100 is prescribed and held constant thereafter. The melting rate of 0.1 Sv is adopted from simulations by interactive ice sheet models for a quadrupling of pre-industrial atmospheric $CO_2$ concentrations (e.g., Mikolajewicz et al., 2007; Vizcaíno et al., 2010) and used in several other studies to investigate GIS melting impacts, e.g., on the strength of the AMOC (Stouffer et al., 2006; Yu et al., 2016; Swingedouw et al., 2015). The assumption of a linear increase is an idealized approach to deal with the uncertainty in the construction of a hydrological sensitivity parameter, often defined as a constant freshwater discharge per degree atmospheric warming

(e.g., Zickfeld et al., 2008; Kuhlbrodt et al., 2009), and has likewise been applied, e.g., in Jungclaus et al. (2006). For this experiment, we have also performed an ensemble of three realizations, branching from ensemble E0 in the year 2006. Thus, E0 and E010 represent the core experiments to investigate the impact of GIS meltwater discharge on the regime shift in the nutrient supply to the NWES.

In experiment E025, we increase the GIS meltwater discharge to 0.25 Sv during the same period as in E010 (2006–2100). This experiment is designed to further stress results from experiment E010 and to gain more insight into the involved physical processes and driving mechanisms. Besides, the higher discharge value accounts for uncertainties in GIS melting rates projected by ice sheet models (Church et al., 2013; Agarwal et al., 2015). The range covered by experiments E010 and E025 is similar to that implicated in past AMOC shutdowns in climate history (e.g., Hemming, 2004; Jongma et al., 2013; Ziemen et al., 2019). E025 can therefore be considered as a high-end GIS melting estimate and sensitivity experiment. A similarly high discharge rate of 0.2 Sv has been considered as upper limit, e.g., by Kwiatkowski et al. (2019).

It shall be shown in Sect. 3.3 that the impact of GIS meltwater discharge on the onset of the regime shift is comparatively weak. In experiment E100, we therefore provoke a more distinct signal by further increasing the meltwater discharge to an extreme value of 1.0 Sv in the year 2100. This high discharge rate is purely motivated by process understanding and exceeds any present estimate of GIS runoff during the 21st century. In fact, given a present-day GIS volume of about $2.9 \times 10^{15}$ m$^3$, it would lead to a complete disintegration of the ice sheet in the first half of the 22nd century, depending on the surface mass balance.

## 2.3 Model evaluation

Numerous other studies have investigated the impact of an additional freshwater hosing to the northern North Atlantic on the ocean circulation and biological productivity in the context of potential future climate change. To evaluate our downscaling simulations, we briefly compare here projected change signals most relevant for our study, i.e., the shoaling of the wintertime MLD, the weakening of the AMOC and the decline of upper-ocean nutrient concentrations, as well as some variables of general interest, such as sea surface temperature (SST) and net primary production.

Maximum (March) MLD in the NE Atlantic simulated by our downscaling model is about 800 m for 1971–2000 (Fig. 3a) and is generally overestimated by about 200 m (also in the main convection sites) compared to the observational climatology by de Boyer Montégut et al. (2004). An overestimation of wintertime MLD and deep water formation has also been found as a common feature in the majority of global Earth system models of CMIP5 generation (Heuzé, 2017). Partly, this is due to the often applied den-

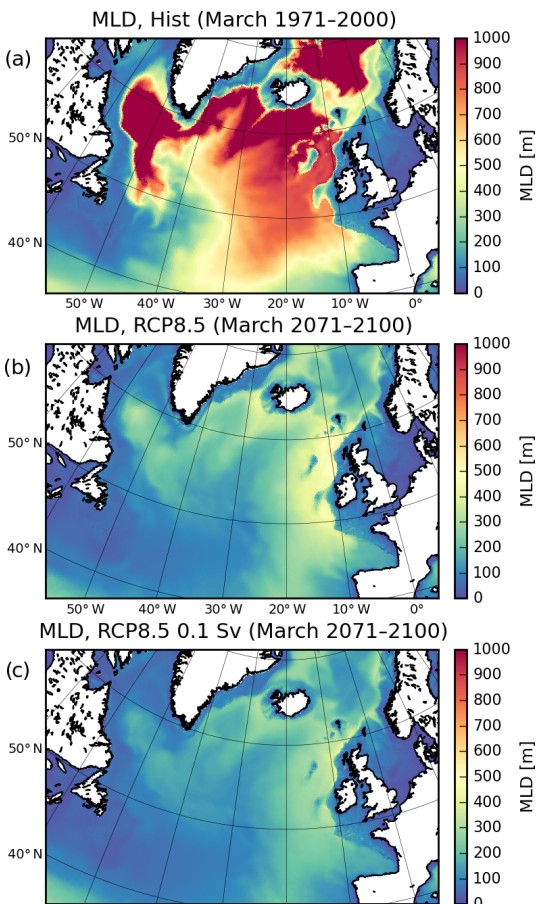

**Figure 3.** Simulated March mixed layer depth in the subpolar North Atlantic under historical conditions (1971–2000) (**a**; experiment E0), RCP8.5 conditions (2071–2100) (**b**; experiment E0) and RCP8.5 conditions (2071–2100) with GIS meltwater discharge of 0.1 Sv (**c;** experiment E010).

sity difference criterion, which leads to comparatively deep MLDs (e.g., Costoya et al., 2014; Courtois et al., 2017). In the present study, we have used a density threshold of $\Delta\sigma_\theta = 0.125$ kg m$^{-3}$ (e.g., Suga et al., 2004).

During the 21st century, NE Atlantic MLD decreases by about 400 m under RCP8.5 conditions without additional GIS meltwater release (E0; Fig. 3b) and by about 500 m with an increasing GIS melting rate of up to 0.1 Sv (E010; Fig. 3c). The MLD changes projected by other studies reflect the wide range of simulated ocean stability across the applied circulation models. In a stronger warming scenario by Kuhlbrodt et al. (2009), North Atlantic MLD decreases but deep water formation does not stop completely (in the Nordic Seas). With a maximum hosing of 0.5 Sv, there is no MLD deeper than 200 m anywhere north of 45° N. These results are similar to our simulations with both significantly weaker radiative forcing and freshwater input.

The present-day strength of the AMOC has been estimated from observations to be around 16–20 Sv (Talley et al.,

2003; Kanzow et al., 2010; Fu et al., 2018) and is well captured by our downscaling model, simulating $17.7 \pm 1.1$ Sv at $30°$ N during 1971–2000. Associated with the range of MLD sensitivity, Earth system models show a large range of AMOC sensitivity to additional freshwater input to the North Atlantic. Accordingly, previous studies can be grouped by their simulated AMOC responses into low freshwater impact ($< 30$ % AMOC weakening within 100 years due to $\sim 0.1$ Sv freshwater hosing), moderate impact ($30$ %–$60$ % AMOC weakening) and high impact ($> 60$ % AMOC weakening). Low impacts have been projected, e.g., by Mikolajewicz et al. (2007); Driesschaert et al. (2007) and Hu et al. (2009), low to moderate impacts (depending on the model), e.g., by Stouffer et al. (2006) and Swingedouw et al. (2013), moderate impacts, e.g., by Jungclaus et al. (2006) and Swingedouw et al. (2015), and high impacts, e.g., by Zickfeld et al. (2008) and Kuhlbrodt et al. (2009). The additional AMOC weakening of about $10$ % in our hosing experiment (E010; see Sect. 3.1) classifies our downscaling model as a low impact model with respect to AMOC sensitivity.

Furthermore, in our simulations, the hosing causes a freshening of the entire subpolar and arctic North Atlantic compared to E0. The freshening of the subpolar region is also consistent among global Earth system models (Swingedouw et al., 2013). Some models, however, show a positive salinity anomaly in the Nordic Seas because more Atlantic waters reach the Arctic. This is also associated with a warming of the Nordic Seas but a cooling everywhere else, which is also indicated in our simulations. With additional freshwater input, the cooling becomes stronger and the warming of the Nordic Seas weaker. Experiments by Kuhlbrodt et al. (2009), by contrast, show amplified cooling in particular in the Nordic Seas.

Similarly, the impact of climate change on marine primary production and upper-ocean nutrient concentrations has considerable spread among global Earth system models (Bopp et al., 2013; Fu et al., 2016). Nevertheless, in the subpolar North Atlantic, the majority of models show a negative response under emission scenario RCP8.5 due to a weakening of the vertical nutrient supply from intermediate depths to the upper ocean. In agreement with these trends, the biogeochemical response in our simulations to a MLD decrease and AMOC weakening is a general reduction in surface nutrient concentrations and net primary production in the NE Atlantic by about $60$ % and $40$ %, respectively (see Sect. 3.1). Comparable changes have been found by Zickfeld et al. (2008) and Kwiatkowski et al. (2019). A rather strong signal of up to $90$ % reduction in primary production was simulated by Kuhlbrodt et al. (2009) due to their stronger warming scenario and higher freshwater input. Nevertheless, similar to our study, the climate-induced changes showed a distinctly stronger impact on primary production than the relative changes induced by freshwater hosing.

It has to be noted, however, that the cited model experiments differ in the precise location, timing and amount of the imposed freshwater hosing. Unless an interactive ice sheet model is involved, the additional freshwater is typically released either uniformly over a wide region in the North Atlantic or restricted to the coast of Greenland with a sudden introduction of a constant discharge rate or a transient increase over several decades or centuries under past, present-day or future climatic conditions. The differences in the experiment design impede direct comparison, often requiring scaling assumptions, and may contribute to the wide spread in the simulated responses of the ocean circulation. After all, with respect to MLD and AMOC perturbations, our downscaling model is comparatively sensitive to changes in the atmospheric forcing and comparatively insensitive to additional freshwater input to the northern North Atlantic.

## 2.4 Analysis

In M19, the transition to the future shallow-ML regime was identified to happen near the end of the 21st century. In the present study, climate change signals are therefore referred to as differences between the distant future period (2101–2150) and the recent past (1971–2000). Time series are shown for 1960–2150, excluding the period 1920–1959 considered as spinup. For our analysis, oceanic nutrients in general are represented by dissolved phosphate concentrations, consistent with M19. Nitrate concentrations, for example, are subject to nitrogen fixation and denitrification processes, making phosphate a more appropriate candidate for the investigation of the impact of climatic changes on oceanic tracer advection.

The bathymetry of the NWES region is shown in Fig. 4. Transect S1 is used to illustrate changes in the vertical nutrient distribution near the shelf break. Budgets for the northern North Sea (Sect. 3.4) are calculated over an area bounded in the north by the solid line (Orkney–Shetland and Shetland–Norway at $60.4°$ N) and in the south by the $50$ m isobath.

## 3 Results

### 3.1 Changes in the mean ocean–shelf nutrient transport

The general freshening and warming of the upper ocean in the subpolar North Atlantic during emission scenario RCP8.5 leads to a decrease in sea surface salinity, a strengthening of the permanent stratification (Fig. A2a, b) and a reduction of the wintertime MLD in the NE Atlantic (Fig. 5a, b). The additional freshwater hosing due to GIS meltwater discharge further intensifies these changes (Figs. 5c, d and A2c, d).

In experiment E0, the shoaling of the wintertime ML results in a limited vertical nutrient supply to the euphotic zone (Fig. 5) and thus weakens the biological productivity during the following summer season (Gröger et al., 2013; Mathis et al., 2018, 2019). As near the end of the 21st century the ML becomes as shallow as the NWES, mixing

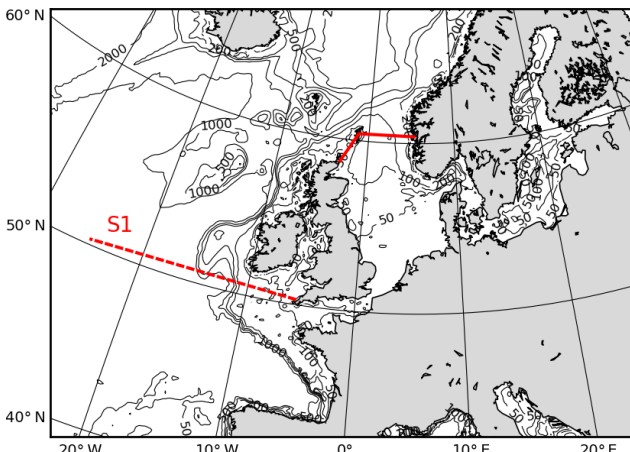

**Figure 4.** Bathymetry of NWES region. S1: transect used to illustrate changes in the vertical nutrient distribution near the shelf break. Solid red line: northern boundary of the area used to calculate budgets for the northern North Sea. The southern boundary is the 50 m isobath.

processes at the shelf break maintain local vertical nutrient fluxes. Nutrient-enriched water masses spread from the shelf break to the interior of the shelf and yield weaker reductions in nutrient concentrations and productivity there (M19). When GIS meltwater is taken into account (experiments E010 and E025), the MLD in the NE Atlantic decreases further and nutrient concentrations in the upper ocean decline more (Fig. 5c, d). Surprisingly though, nutrient concentrations on the shelf decrease even less than in the experiment without GIS meltwater discharge (E0). The regime shift in E0 is characterized by a disruption of the ocean–shelf signal transfer by mixing processes at the shelf break. The contrasting impact on the nutrient distribution seen in the meltwater discharge experiments (E010 and E025) indicates that also here the change signal in the NE Atlantic is not coherently transferred to the shelf but there are other mechanisms involved influencing the on-shelf nutrient transport.

These opposite effects in the open ocean and shallow shelf are a consequence of the effect of the GIS meltwater on the North Atlantic circulation at a basin-wide scale. In E0, the climate change signal from the atmosphere (surface warming and increased moisture transport) drives a general slowdown of the AMOC (Fig. 6a). The resulting weakening of the northward heat transport enhances the meridional temperature gradient in the ocean and atmosphere and induces a northward shift of the atmospheric wind field, as indicated in most CMIP3 and CMIP5 models (e.g., Yin, 2005; Woollings et al., 2012; Hu et al., 2013; Fischer et al., 2017). The associated strengthening of the westerlies at midlatitudes by up to 30 % and the increase in wind stress curl over the NE Atlantic by up to 15 % drive an expansion of the subpolar gyre (SPG) towards the east (Fig. 7). Moreover, in the western part of the

gyre system, the boundary between the SPG and subtropical gyre moves northward.

In the upper ocean, the widening of the SPG leads to a reduction in the northward intrusion of saline subtropical water to the NE Atlantic and to a stronger eastward propagation of low-salinity subpolar water. The related decrease in the density strengthens the stratification and reduces the MLD in addition to the climate change signal from the atmosphere. Accordingly, GIS meltwater discharge causes a stronger salinity drop in the SPG region and hence amplifies the impact on the stratification and MLD (Fig. 5c, d). The resulting weakening of the vertical nutrient supply from the ocean's interior leads to lower nutrient concentrations in the NE Atlantic wintertime ML when more GIS meltwater is added.

The additional source for the nutrients on the NWES (Fig. 5c, d), by contrast, is found in Atlantic water masses below the pycnocline. The weakening of the AMOC (Fig. 6a) is associated with a general weakening of the North Atlantic deep circulation. Deep water formation in the Labrador Sea, Irminger Sea and Nordic Seas CE3 breaks down around 2080 in all experiments. Subpycnocline water masses thus become older with respect to the time since they were last in contact with the atmosphere. Continuous remineralization of sinking particulate organic matter is at the expense of dissolved oxygen (e.g., Keeling et al., 2010) but releases nutrients (Fig. 6b, c). When GIS meltwater is added, the density of the upper ocean decreases more CE4, leading to a slightly stronger slowdown of the AMOC (Fig. 6a). Thus, more organic material is remineralized, enhancing subpycnocline oxygen utilization and increasing nutrient concentrations. During the future shallow-ML regime, finally, the nutrient-enriched subpycnocline water masses are mixed to the upper water column near the shelf break and spread over the shelf, causing elevated nutrient concentrations there.

Ocean–shelf exchange is mainly governed by wind-driven Ekman transport, internal tidal waves and cross-shelf break transport induced by the northward-flowing slope current (e.g., Huthnance et al., 2009; Simpson and Sharples, 2012; Ruiz-Castillo et al., 2019). The simulated vertical profile of cross-shelf break transport during winter (DJFM) integrated along the 200 m isobath between 47 and 61.5° N (Fig. 8a) shows the typical structure of a strong wind-driven on-shelf surface flow (0–40 m), an on-shelf flow in the interior of the water column (40–130 m) and an off-shelf Ekman drain at the bottom (130–200 m) due to basal stress of the slope current (Graham et al., 2018). Independent of the GIS melting rates, the net on-shelf winter transport through this section weakens under RCP8.5 by about 23 % (from 1.11 in 1971–2000 to 0.86 Sv in 2101–2150). Most prominent changes are indicated in the on-shelf interior and off-shelf bottom transports (Fig. 8a), induced by a weakening of the slope current (Fig. 9a).

The corresponding phosphate ($PO_4$) profiles at the shelf edge (Fig. 8b) illustrate the projected general nutrient decline in the upper water column as well as the influence of

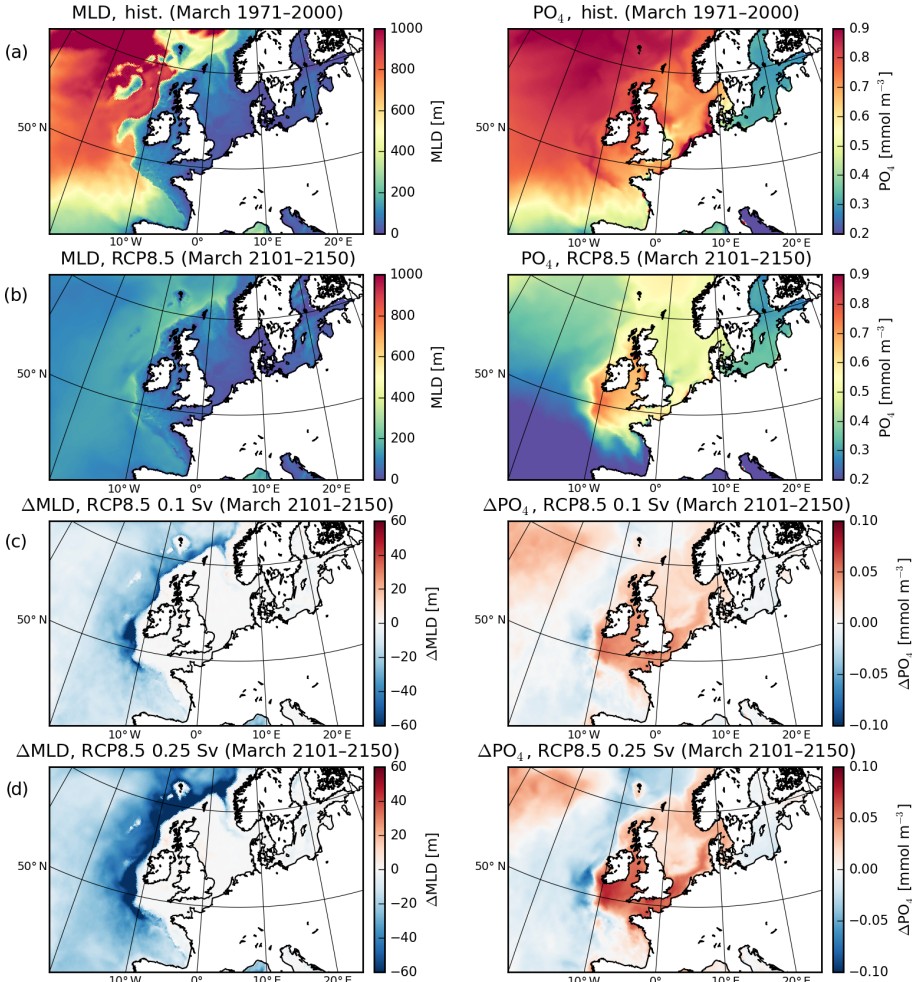

**Figure 5.** Maximum wintertime (March) MLD and pre-bloom (March) surface PO$_4$ concentration for historical (1971–2000, **a**) and RCP8.5 (2101–2150, E0, **b**) conditions. Changes in RCP8.5 conditions (2101–2150) for experiments with 0.1 Sv (E010, **c**) and 0.25 Sv (E025, **d**) GIS meltwater discharge relative to the experiment without GIS meltwater discharge (E0, **b**).

the increasing nutrient concentrations at deeper levels. When GIS meltwater is added, the nutrient enrichment of subpycnocline Atlantic water masses results in a relative increase of future nutrient concentrations near the shelf break, mitigating the projected nutrient decline on the shelf. Net on-shelf PO$_4$ fluxes decrease from 839 (1971–2000) to 311 mol s$^{-1}$ (2101–2150) in E0, 265 in E010 and 211 mol s$^{-1}$ in E025. This increasing reduction among the experiments, however, is dominated by the decreasing volume transports ($-0.25$ Sv in E0, $-0.30$ in E010, $-0.32$ in E025) and does not reflect the changes in the nutrient concentrations. As future mean pre-bloom PO$_4$ concentrations on the shelf range between 0.55 and 0.7 mmol m$^{-3}$ in E0, with maximum values of 0.8 mmol m$^{-3}$ in E025 (Fig. 5b–d), the on-shelf transport in the interior of the water column is an essential feature to establish these elevated concentrations. It is therefore important to understand the weakening of the slope current and its impact on the cross-shelf break exchange.

Primarily, the northward-flowing slope current along the continental margin of the NWES is driven by the meridional density gradient at intermediate depths to the west of the shelf break (Marsh et al., 2017). The region of influence has been identified to span from about 45 to 62° N and zonally as far as 28° W. The meridional density gradient in this region induces a geostrophic flow towards the east which turns northward as it approaches the shelf break, forming the core of the slope current. Under RCP8.5 conditions (E0), the meridional density difference (between north and south) decreases from about 0.42 (1971–2000) to 0.27 kg m$^{-3}$ (2101–2150; Fig. 9b and Table 2). The slowdown of the AMOC and the expansion of the SPG entail an eastward migration of old Labrador Sea water and water of Arctic and Nordic Seas origin flowing through the Denmark Strait (Fig. A3) as well as a reduction of the northward migration of warm and saline subtropical water, leading to a freshening and cooling of NE Atlantic intermediate waters (Fig. 10). A stronger impact on

**Table 2.** [TS1] Annual mean meridional density difference ($\rho$ diff) in the NE Atlantic (45–62° N, 15–28° W) averaged over 500–1000 m depth as well as change (2101–2150 minus 1971–2000) in temperature ($T$), salinity ($S$) and density ($\rho$) in the north (subscript N) and south (subscript S) of the given NE Atlantic region. Ensemble spread for E0 and E010 is about ±10 %.

| Experiment | $\rho$ grad. (kg m$^{-3}$) | $\Delta T_N$ (K) | $\Delta T_S$ (K) | $\Delta S_N$ (psu) | $\Delta S_S$ (psu) | $\Delta \rho_N$ (kg m$^{-3}$) | $\Delta \rho_S$ (kg m$^{-3}$) |
|---|---|---|---|---|---|---|---|
| E0 (1971–2000) | 0.42 | – | – | – | – | – | – |
| E0 (2101–2150) | 0.27 | +1.00 | −0.05 | −0.28 | −0.24 | −0.34 | −0.19 |
| E010 (2101–2150) | 0.27 | +0.76 | −0.40 | −0.34 | −0.33 | −0.35 | −0.20 |
| E025 (2101–2150) | 0.26 | +0.61 | −0.90 | −0.36 | −0.43 | −0.34 | −0.19 |

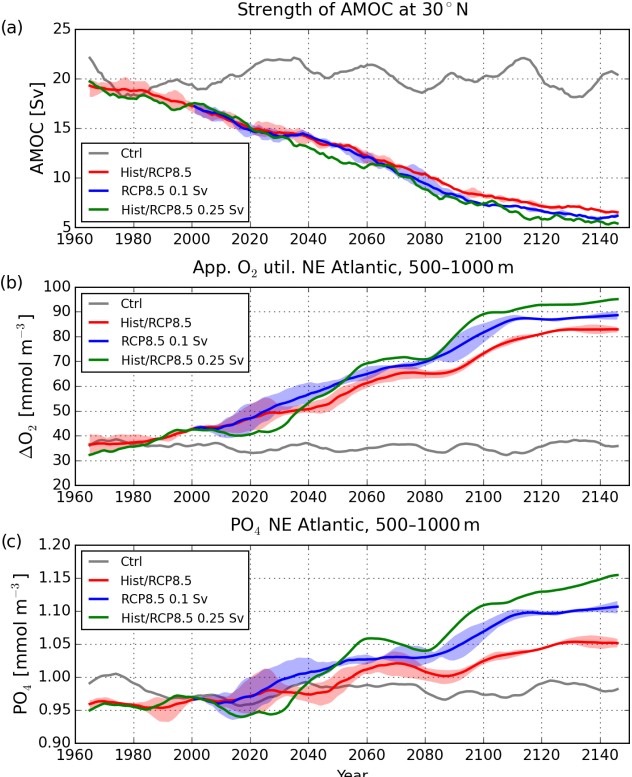

**Figure 6.** Strength of Atlantic meridional overturning circulation **(a)**, represented by the vertical maximum of the annual mean stream function of the zonally integrated meridional transport at 30° N. Annual apparent oxygen utilization **(b)** and phosphate concentration **(c)** in the NE Atlantic (45–62° N, 15–28° W) at 500–1000 m depth. Solid: control run C0 (gray), the ensemble means of experiments E0 (red) and E010 (blue), and a single realization of E025 (green). Shaded areas: ensemble spreads.

the density though is indicated in the northern region of influence (−0.34 kg m$^{-3}$; Table 2), where the cooling effect is weakest and the water temperature thus still increases due to the increasing radiative forcing. In the south, the remaining influence of warm subtropical water and Mediterranean overflow water induces a more moderate change (−0.19 kg m$^{-3}$). Because of the stronger density drop in the north than in the south, the meridional density gradient decreases.

As GIS meltwater is added, both the freshening and cooling of the NE Atlantic intermediate water become stronger but their effects on the water density roughly cancel (Table 2). The decrease of the meridional density gradient is therefore very similar for all experiments with and without meltwater discharge. Moreover, most significant changes happen during the period when the SPG widens (Fig. 11). Around the year 2080, the SPG reaches its maximum size, limited by the topographic boundary of the European continent (Fig. 7). After 2080, the meridional density gradient as well as the core of the slope current do not change further (Fig. 9), resulting in similar subsurface transport profiles among the experiments during the new shallow-ML regime (Fig. 8a).

Regarding the nutrient fluxes to the shelf, though, the weakening of the interior on-shelf transport from historical to future conditions is partly compensated by the influence of nutrient-rich subpycnocline water masses mixed up at the shelf break. When GIS meltwater is taken into account, subpycnocline nutrient concentrations become even higher and thus lead to an intensification of the developed ocean–shelf nutrient front (Fig. 5c, d).

### 3.2 Changes in the variability of ocean–shelf nutrient transport

The interannual to multidecadal variability of the advective nutrient supply to the NWES has been shown to increase rapidly during the shallow-ML regime (M19). The main mechanism behind this is enhanced MLD variations near the shelf break which control the nutrient concentration of the on-shelf transport along the continental margin. The strongest signal though was found at the Celtic shelf break. This feature is confirmed by our additional RCP8.5 realizations (Figs. 12a and A5a), showing an increase in the standard deviation of surface PO$_4$ concentrations at the Celtic shelf break (Fig. 4, S1) by a factor of 4.4, from 0.027 (1971–2000) to 0.120 mmol m$^{-3}$ (2101–2150, E0).

We find corresponding anomalies in the future sea level pressure (SLP) field to be related to variations in the strength and orientation of the SLP gradient between the Iceland Low and Azores High (Fig. 13). The drivers for the atmospheric anomalies are sporadic southeastward displacements of the

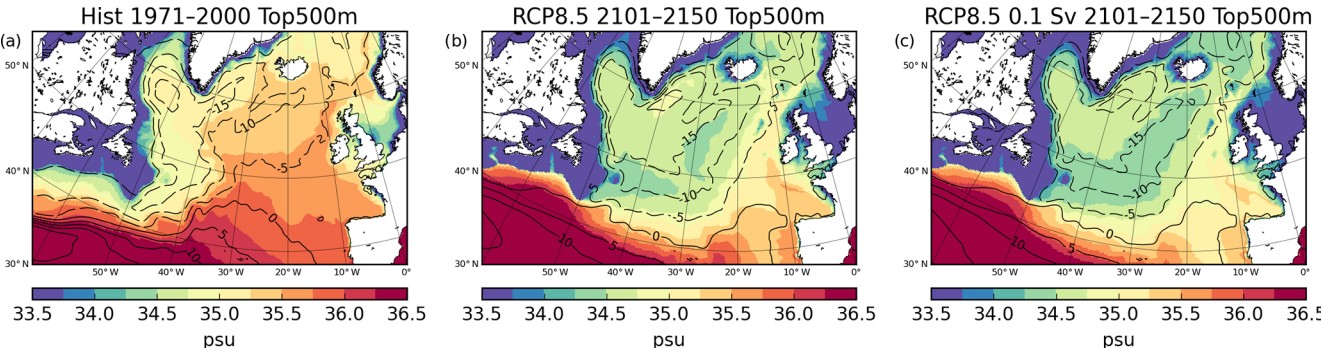

**Figure 7.** The geometry of the subpolar gyre for historical conditions (1971–2000, **a**) and RCP8.5 conditions (2101–2150) for experiments without (E0, **b**) and with (E010, **c**) additional GIS meltwater discharge. Contour lines: annual mean stream function of the vertically integrated transport in the upper 500 m. Colors: sea surface salinity.

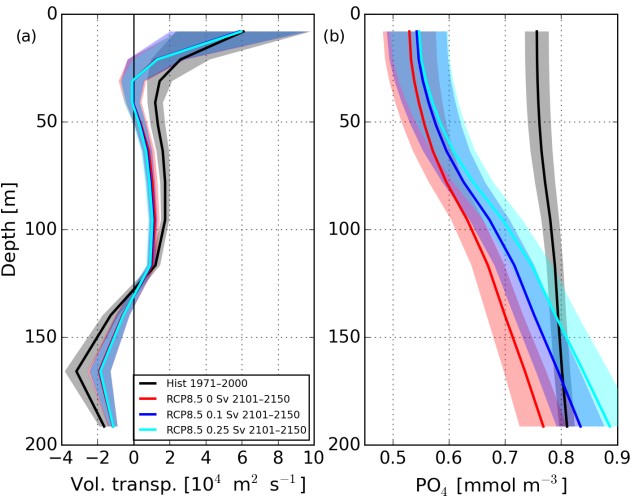

**Figure 8.** Profile of cross-shelf break transport (positive on-shelf, negative off-shelf) during winter (DJFM) integrated along the 200 m isobath from 47 to 61.5° N (**a**). Corresponding profile of mean phosphate concentration (**b**). Solid: ensemble means of historical conditions (1971–2000, black) and RCP8.5 conditions (2101–2150) for experiments E0 (red) and E010 (blue), as well as a single realization of E025 (cyan). Shaded areas: ensemble means of standard deviation (and standard deviation of E025).

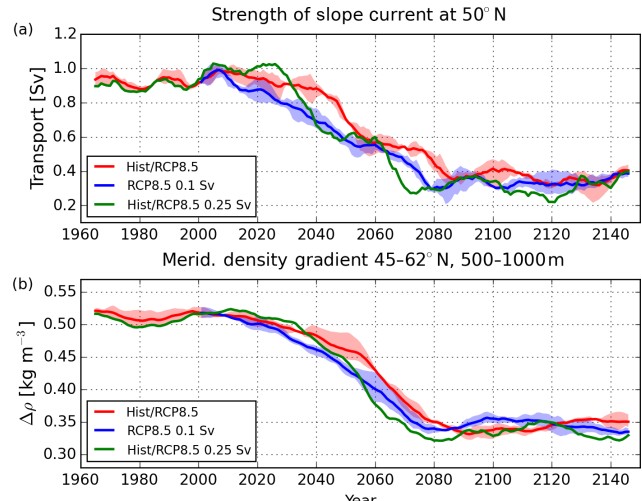

**Figure 9.** Strength of northward-flowing slope current during winter (DJFM) at 50° N (**a**). Annual meridional density gradient in the NE Atlantic (45–62° N, 15–28° W) at 500–1000 m depth (**b**). Solid: control run C0 (gray), the ensemble means of experiments E0 (red) and E010 (blue), and a single realization of E025 (green). Shaded areas: ensemble spreads.

Iceland Low, not represented by any of the leading variability modes of North Atlantic SLP (not shown). The resulting SLP anomalies over the NE Atlantic cause near-surface variations in the meridional wind component, the wind speed and air temperature, affecting the surface heat flux between the ocean and atmosphere and thus the ocean MLD (Fig. A4). In E0, the variability of the atmospheric forcing does not significantly increase through the scenario period. During the shallow-ML regime, however, upper-ocean conditions are more sensitive to variations in the atmospheric forcing. A positive SLP anomaly is associated with southward advection of cold Arctic air masses and enhanced wind speeds. The enhanced surface cooling promotes convection in the upper

ocean and deepens the ML. In particular, near the shelf break, the deepened ML entrains warm and saline subpycnocline water to the upper ocean and leads to positive SST and sea surface salinity (SSS) anomalies. Part of the anomalous heat is released to the atmosphere while the salt remains in the water, which increases the upper-ocean density. This mechanism has been shown to feed back to the vertical convection and MLD (M19).

When GIS meltwater is taken into account (E010 and E025), there is a weak increase in the future variability of the atmospheric forcing over the NE Atlantic (Table 3). Moreover, the mean wintertime MLD in the NE Atlantic is more shallow than in E0 (Fig. 5c, d), enhancing the influence of the atmospheric forcing on the MLD (Table 3). However, the

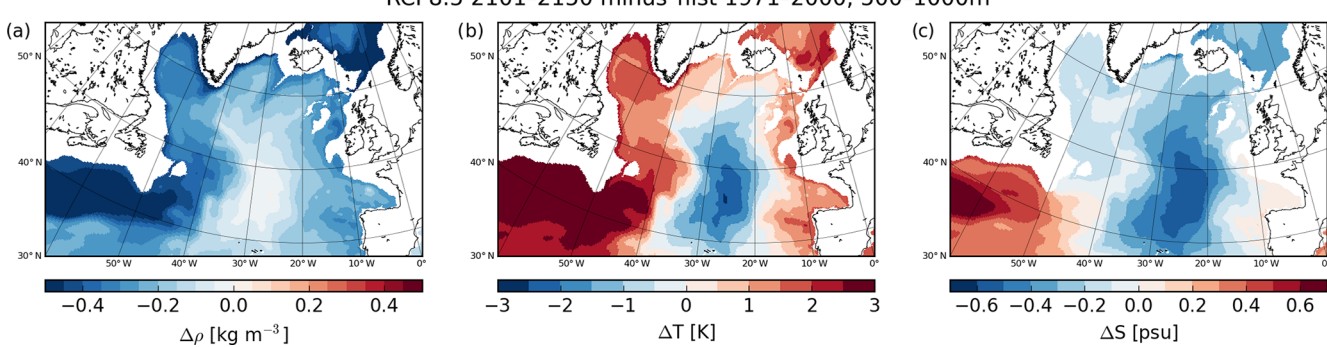

**Figure 10.** Ensemble means of projected changes (RCP8.5 2101–2150 E0 minus historical 1971–2000) in annual means of seawater density **(a)**, temperature **(b)** and salinity **(c)** at intermediate depths (500–1000 m).

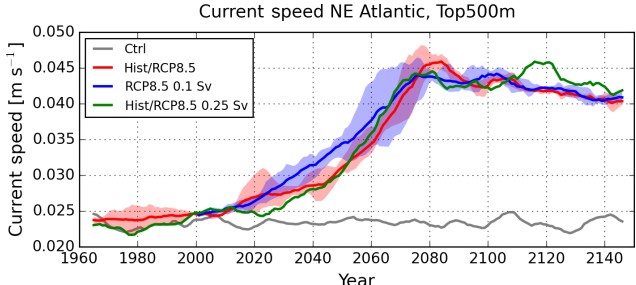

**Figure 11.** Mean current speed of the upper 500 m in the NE Atlantic (45–62° N, 15–28° W). Solid: control run C0 (gray) and ensemble means of experiments E0 (red), E010 (blue) and E025 (green). Shaded areas: ensemble spreads.

impact on the variability of upper-ocean nutrient concentrations is very weak. The main impact is due to the stronger vertical nutrient gradient, resulting from the shallower MLD and higher subpycnocline nutrient concentrations (Sect. 3.1). The stronger nutrient gradient translates MLD variations to enhanced variations in the vertical nutrient supply. Accordingly, for the shallow-ML regime, the standard deviation of surface $PO_4$ concentrations at the Celtic shelf break increases further to 0.148 mmol m$^{-3}$ for E010 (Fig. 12b) and 0.153 mmol m$^{-3}$ for E025 (Fig. 12c). Comparing the standard deviations given here and in Table 3, the increase in the variability of upper-ocean nutrient concentrations by 23 % for E010 (compared to E0) decomposes into contributions of only 2 % from the atmospheric forcing and 21 % from the vertical nutrient gradient.

The described mechanisms similarly affect the vertical distribution of salinity in the open ocean as well as the variability of the on-shelf salt flux (Fig. A7). In the NE Atlantic, the freshening of the upper ocean under RCP8.5 leads to the development of a vertical salinity gradient through the pycnocline of 1.20 psu (2101–2150) in E0. In E010 and E025, this gradient strengthens further to 1.65 and 1.94 psu, respectively, due to the additional freshwater hosing at the sea sur-

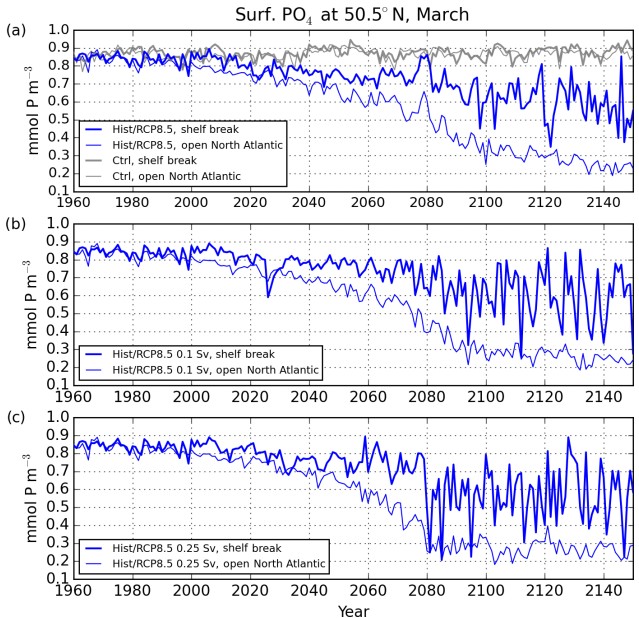

**Figure 12.** Pre-bloom (March) surface phosphate concentrations at 50.5° N (Fig. 4, S1) at the Celtic shelf break (10–12° W, thick lines) and in the open NE Atlantic (15–20° W, thin lines) for the control run C0 (gray) and single realizations (blue) of experiments E0 **(a)**, E010 **(b)** and E025 **(c)**. Two more ensemble members for E0 and E010 are shown in Fig. A5.

face. Moreover, in all experiments, the mixing of saline subpycnocline water masses to the upper ocean near the shelf break establishes an ocean–shelf salinity front with higher SSS in the Celtic Sea than in the NE Atlantic (Fig. A2). Finally, the variability of SSS at the Celtic shelf break increases substantially from 0.03 psu (standard deviation 1971–2000) to 0.25 psu (2101–2150) in E0 and further to 0.33 in E010 and 0.36 psu in E025. Unlike nutrient concentrations, however, the stronger increase in the variability of salinity in E010 and E025 is induced by the stronger change (freshening) of the upper ocean. Subpycnocline salinities rather show

**Table 3.** Variation in the wintertime (DJFM) atmospheric forcing over the NE Atlantic during the shallow-ML regime as the standard deviation of the projection of the SLP composite pattern (Fig. 13) on the unprocessed SLP time series ($\sigma PC_{SLP}$) and the ratio between the variability and the mean of the March MLD at the Celtic shelf break ($\sigma MLD$/mean MLD). Ensemble spread for E0 and E010 is about $\pm 10\%$. The relation between SLP and MLD anomalies is explained in Sect. 3.2.

| Experiment | $\sigma PC_{SLP}$ | $\sigma MLD$/ mean MLD |
|---|---|---|
| E0 (2101–2150) | 1.05 | 0.33 |
| E010 (2101–2150) | 1.07 | 0.39 |
| E025 (2101–2150) | 1.10 | 0.42 |

a weak decrease the more GIS meltwater is added, which is discussed in the next section.

## 3.3 Changes in the timing of the ocean–shelf exchange regime shift

The shallow-ML regime is fully established when the ML near the shelf break becomes as shallow as the depth of the shelf edge, i.e., about 150–200 m. As can be expected, the impact of GIS meltwater discharge on the stratification strengthening and ML shoaling (Sect. 3.1) implies that the MLD in the NE Atlantic decreases faster when GIS meltwater is added, passing this critical depth earlier in the 21st century (Fig. A6). Nevertheless, the changes in the regime shift timing are surprisingly weak, given the melting rates vary considerably between the experiments. For a melting rate of 0.1 Sv (E010), the impact is only about 10 years (see also PO$_4$ time series shown in Figs. 12 and 16).

The freshwater hosing to the upper ocean leads to a reduction in upper-ocean salinity (Fig. A2c, d). Besides the freshening of the upper ocean, however, the salinity of subpycnocline water masses shows a slight decrease too (Fig. A7). The weakening of the AMOC reduces the northward migration of saline subtropical water and the SPG expansion fosters eastward migration of fresh Labrador Sea and Arctic waters at intermediate depths. This subpycnocline freshening weakens the GIS meltwater impact on the stratification strengthening and ML shoaling, imposing a damping effect on the regime shift timing. Even in a hypothetical extreme scenario of 1.0 Sv meltwater discharge (E100), the shallow-ML regime does not become initiated earlier than the 2070s (Fig. A5d).

## 3.4 Impact on the biological productivity of the northern North Sea

In M19, the projected changes in the Atlantic nutrient supply to the NWES have been shown to impact the biological productivity in the northern North Sea, as being the largest shelf area primarily influenced by Atlantic inflow. The im-

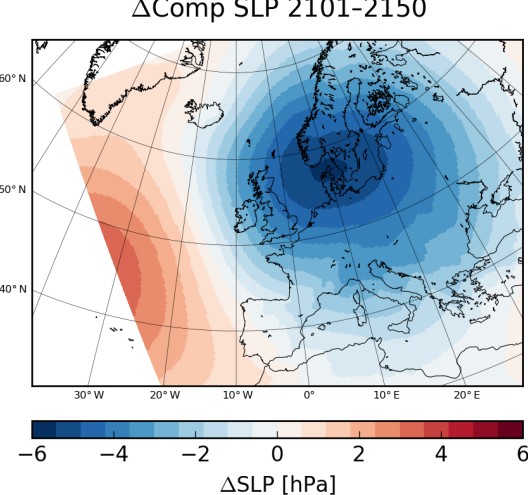

$\Delta$Comp SLP 2101–2150

$\Delta$SLP [hPa]

**Figure 13.** Ensemble mean over seven simulations (E0, E010, E025) of the difference between positive and negative composites of winter (DJFM) sea level pressure during the shallow-ML regime. The years from which the composites are calculated are chosen from the phosphate time series at the Celtic shelf break (Fig. 12), i.e., when phosphate concentrations exceed $\pm 1$ standard deviation.

port of nutrient-rich subpycnocline water masses mixed up at the shelf break mitigates the general weakening in the northern North Sea primary production to be expected from the reduction of upper-ocean nutrient concentrations in the NE Atlantic. Nevertheless, the enhanced variability in the on-shelf nutrient transport also leads to enhanced variations in primary production, as the local influence of the strength of the seasonal stratification decreases to a subordinate factor. In this subsection, we therefore investigate the impact of GIS meltwater discharge on the Atlantic nutrient import and net primary production in the northern North Sea.

A more detailed analysis of the driving mechanisms of the slope current reveals that subtle changes in the northern part of the slope current affect the inflow of Atlantic water masses to the northern North Sea (Fig. 14). Until about the year 2080, the warming and freshening of the upper ocean due to the climate change signal in the atmospheric forcing are stronger on the shallow NWES than in the open NE Atlantic. The ocean–shelf gradient in density and sea surface height (SSH) increases and induces a geostrophic flow along the slope (Simpson and Sharples, 2012; Marsh et al., 2017). Moreover, the westerly winds become stronger over the northern NWES region (Sect. 3.1). The wind component parallel to the shelf break increases by 20 %–30 % and thus contributes further to a temporary strengthening of the slope current. During this period, the enhancing effects of the SSH gradient and wind forcing outweigh the weakening effect of the meridional density gradient (Sect. 3.1), leading to an increase in the Fair Isle inflow to the North Sea and no change in the East Shetland inflow (Fig. 14a, b). These

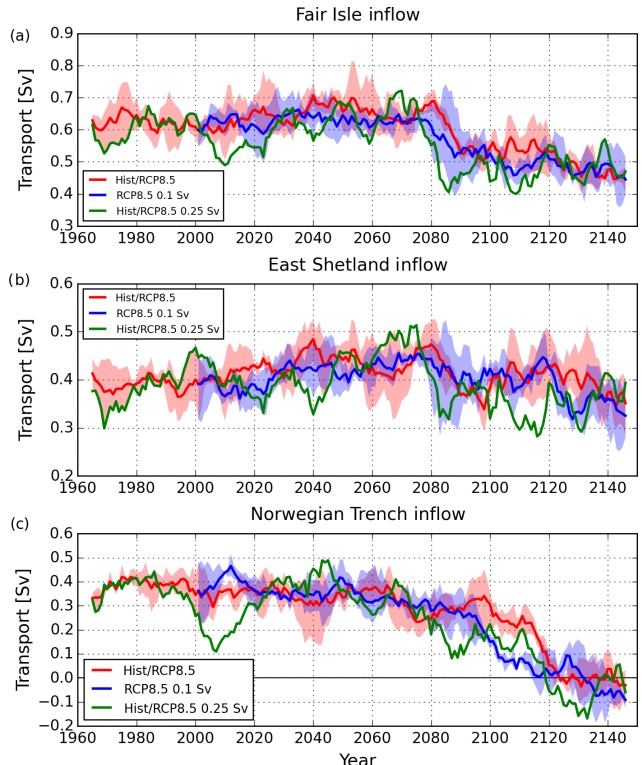

**Figure 14.** Volume transport during winter (DJFM) of the main Atlantic inflow branches to the northern North Sea: Fair Isle Current (**a**), East Shetland flow (**b**), inflow along the western side of the Norwegian Trench at 60.4° N (**c**). Solid: ensemble means of experiments E0 (red) and E010 (blue) and a single realization of experiment E025 (green). Shaded areas: ensemble spread.

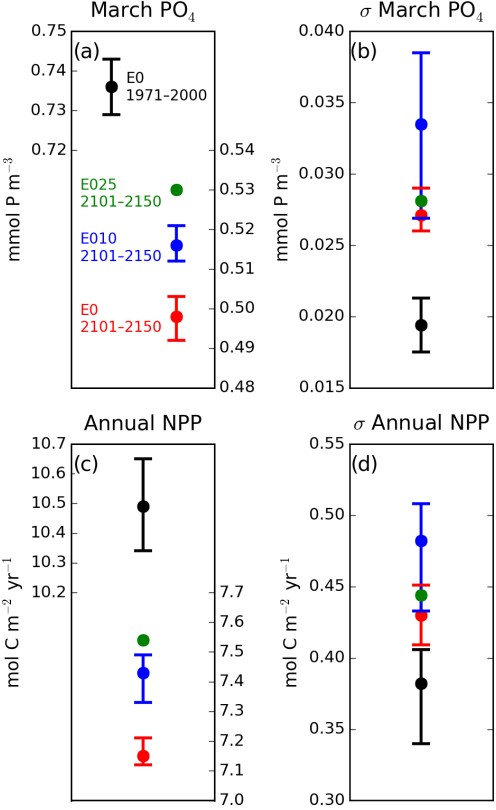

**Figure 15.** Mean (**a, c**) and standard deviation (**b, d**) of pre-bloom (March) phosphate concentration (**a, b**) and annual net primary production (**c, d**) in the northern North Sea (see Fig. 4) during historical conditions (E0, black) as well as the shallow-ML regime for experiments without (E0, red) and with 0.1 Sv (E010, blue) and 0.25 Sv (E025, green) GIS meltwater discharge. For E0 and E010, circle markers represent ensemble means; error bars illustrate ensemble spreads as the range between the minimum and maximum ensemble members. Exact values are given in Table A1.

inflows are further maintained by the enhanced wind-driven Ekman transport.

Around 2080, the expansion of the SPG reaches the continental margin and causes sudden decreases in the ocean–shelf gradients of salinity (by 0.22 psu), density (by 0.17 kg m$^{-3}$) and SSH (by 0.08 m) within one decade. After this period, the weak meridional density gradient west of the shelf break (Fig. 9b) dominates the slope current also in the northern region, and the Fair Isle and East Shetland inflows start to weaken as well.

The inflow along the western side of the Norwegian Trench is mainly governed by topographic steering as the slope current follows the sharp topographic turn to the right at about 62° N. Our simulations indicate a substantial weakening of this inflow (Fig. 14c) and are in line with the findings by Holt et al. (2018). The proposed driving mechanism is an increase in the deformation radius due to the strengthening of the stratification. In addition to the general weakening of the slope current, a smaller fraction of the slope current is then able to follow the topographic turn. In our simulations, the permanent stratification around the entrance of the Norwegian Trench strengthens by about 5 g m$^{-4}$ in experiment

E0 and 7 g m$^{-4}$ in E010 (Fig. A2), and the baroclinic deformation radius increases by about 4–5 km in E0 (similar to 3–4 km in Holt et al., 2018) and 6–7 km in E010. The maximum deformation radii at the end of the simulations are about 10 and 12 km, respectively. During the shallow-ML regime (2101–2150), the correlation between the Norwegian Trench inflow and the deformation radius is about −0.35 (detrended time series) for both experiments. After around 2120, the remaining inflow to the Norwegian Trench does not penetrate further south than about 60° N, implying a reduction down to 0 in Fig. 14c.

In Holt et al. (2018), the changes in the stratification and deformation radius are amplified by a strengthening of the circulation in the Nordic Seas, in particular the East Iceland Current, leading to a southward intrusion of cold and fresh Arctic water masses into the northern inflow region of the North Sea. Similar to that in CE5 our simulations, the Nordic Seas circulation strengthens and a larger fraction of the East

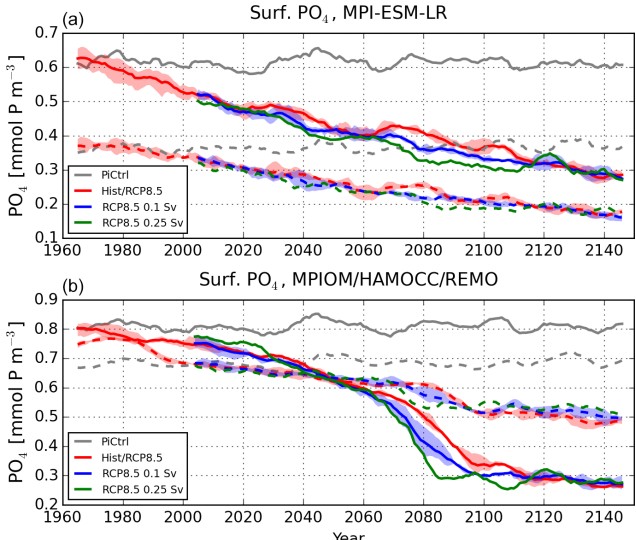

**Figure 16.** Pre-bloom (March) surface phosphate concentration in the NE Atlantic (solid; 45–62° N, 5–20° W) and the northern North Sea (dashed; northern boundary: Orkney–Shetland and Shetland–Norway at 60.4° N; southern boundary: 50 m isobath), simulated by the parent global model MPI-ESM-LR **(a)** and the downscaling model system MPIOM/HAMOCC/REMO **(b)**. Lines: control run C0 (gray), ensemble means of experiments E0 (red) and E010 (blue), and single realization of experiment E025 (green). Shaded areas: ensemble spread.

Iceland Current passes the Faroe Islands southeastward, joining the slope current to the north of the Shetland Islands (Fig. A8). Northeast of the Faroe Islands, the East Iceland Current intensifies by about 30 % (in E0 and E010). We thus confirm the enhanced influence of Arctic water masses on the Norwegian Trench inflow projected by Holt et al. (2018). A sudden shutdown due to positive feedback with the accumulation of coastal North Sea water of low salinity, however, does not occur in our simulations. The changes in the circulation are rather smooth over several decades (see Fig. 14c for the Norwegian Trench inflow; not shown for the East Iceland Current) and simulated to happen later than in Holt et al. (2018) by about 30 years. Moreover, a reversal of the Norwegian Trench inflow to a persistent outflow, as also projected by Tinker et al. (2016), is not indicated but may happen after 2150.

Since the future changes in the NE Atlantic circulation and density structure are rather similar in all experiments with and without GIS meltwater discharge, the impact of the freshwater hosing on the slope current and transport rates to the NWES is very low. The main mechanism for the increase in nutrient concentrations on the shelf when GIS meltwater is added thus is the increase in nutrient concentrations of subpycnocline Atlantic water masses (Sect. 3.1). It has been shown by M19 that during the shallow-ML regime the sustained connection to the Atlantic subpycnocline nutrient pool

promotes primary production on the NWES. In our new simulations, accordingly, the further increase in subpycnocline nutrient concentrations as a response to added GIS meltwater discharge leads to an increase in pre-bloom nutrient concentrations in the northern North Sea by about 4 % for E010 and 6 % for E025 (relative to E0 2101–2150) and hence to higher net primary production (NPP; Fig. 15a, c). The projected decrease in northern North Sea productivity during the scenario period by 32 % for E0 (2101–2150 minus 1971–2000) therefore weakens to 29 % for E010 and to 28 % for E025. Similarly, the increase in the variability of Atlantic nutrient supply tied to added GIS meltwater leads to higher variability in both pre-bloom nutrient concentrations and net primary production by up to 16 % under the shallow-ML regime (relative to E0 2101–2150) with considerable uncertainty due to internal variability (Fig. 15b, d). These results demonstrate how under decreasing wintertime MLDs the signal transfer from the open ocean to the shelf becomes more sensitive to climate-induced large-scale changes in the North Atlantic mean circulation and interannual variability.

## 4 Discussion

Our simulations indicate a moderate reduction of net on-shelf transport from the NE Atlantic to the NWES under future RCP8.5 conditions. Considering the net winter transport across the continental margin from 44 to 61.5° N (excluding the Norwegian Trench inflow), the reduction of the on-shelf component from about 3.12 (1971–2000) to 2.47 Sv (2101–2150) reduces the flushing of the open shelf areas during one winter season from about 45 % to 35 %, fairly independent of the considered amount of GIS meltwater. Nevertheless, even though future on-shelf transport weakens, nutrient concentrations on the NWES do not decline as strongly as in the upper NE Atlantic, owing to mixing processes at the shelf break (in agreement with M19). The maintained connection to nutrient-rich Atlantic subpycnocline water masses gains particular importance as the southward turn of the slope current into the Norwegian Trench is found to weaken substantially during the first half of the 22nd century, thus closing the only direct inflow of deeper Atlantic water to the NWES (in qualitative agreement with Holt et al., 2018). As an implication of the weaker circulation, coastal areas in the southern and eastern North Sea become more influenced by river runoff and nutrient loads. In addition to the decrease in the nutrient transport across the shelf break, this further intensifies the mean nutrient gradient between the inner (dominated by river loads) and outer (dominated by Atlantic inflow) shelf areas as well as its interannual variability. For instance, in our experiments, both the mean pre-bloom nutrient gradient between the northern North Sea and the German Bight and its standard deviation increase by about 20 % during the scenario period (2101–2150 minus 1971–2000).

When GIS meltwater is added, our simulations suggest that the projected nutrient decline in the open shelf areas becomes even weaker due to an increase in Atlantic subpycnocline nutrient concentrations. This increase is induced by the stronger slowdown of the North Atlantic deep circulation as a consequence of a stronger AMOC weakening. Moreover, the subpycnocline nutrient enrichment may be underestimated because our model system does not account for the effect of biologically relevant substances transported into the ocean by meltwater and iceberg calving due to microbial activity and hydrolysis reactions at the interface between land ice and the bedrock (such as dissolved iron, silicate and nitrogen; Bhatia et al., 2013; Duprat et al., 2016; Wadham et al., 2016; Hatton et al., 2019). Part of this additional nutrient input to the upper ocean would be consumed by primary producers and exported to deeper levels. A direct connection of NWES water to Atlantic subpycnocline water is established if MLDs are less than about 130 m, which is the maximum depth of the mean on-shelf transport (Fig. 8a). The MLD passes this threshold only in the most extreme experiment of 1.0 Sv meltwater discharge (E100; Fig. A6d). Accordingly, NWES nutrient concentrations in this experiment show the strongest increase by up to 0.13 mmol P m$^{-3}$ compared to the reference RCP8.5 simulation without GIS meltwater discharge (E0).

Furthermore, during the shallow-ML regime, the high variability in on-shelf wintertime nutrient fluxes enhances further when GIS meltwater is added. This results mainly from an increase in the vertical nutrient gradient through the pycnocline and is reinforced by a stronger impact of variations in the atmospheric forcing on the shallower ML. In the upper 30 m of the water column, variations in nutrient concentrations at the shelf break are negatively correlated with the on-shelf transport (about −0.5 for E0 during 2101–2150) mediated by the wind-driven advection of nutrient-poor Atlantic surface water. In the deeper water column (> 30 m), by contrast, this correlation is positive (about +0.4) and increases when GIS meltwater is added (to about +0.6 for E010). Thus, the increasing variability in the nutrient concentrations implies an increasing mean nutrient flux to the shelf via the on-shelf transport in the interior water column (Sect. 3.1). In this way, the enhanced variability in the nutrient concentrations at the shelf break contributes to the elevated mean concentrations on the shelf, in addition to the nutrient enrichment of Atlantic subpycnocline water masses. The contribution to the mean nutrient flux (Sect. 3.1) though is only about 3 % for E010.

The SLP anomaly pattern associated with nutrient anomalies at the shelf break is best represented by a non-stationary combination of the North Atlantic Oscillation (NAO) and Scandinavian (SCAN) teleconnection patterns (Chafik et al., 2017). It also resembles the European–North Atlantic sector of the East Atlantic–west Russian pattern, a higher-order atmospheric mode of the Northern Hemisphere which occasionally mixes with the NAO (Barnston and Livezey, 1987;

Fagherazzi et al., 2005). The other two characteristic pressure centers over west Siberia and China, however, are not shown in the larger SLP anomaly structure corresponding to Fig. 13 (simulated by MPI-ESM; not shown).

While the strength of the NAO was a major factor in the single realization analyzed by M19, we find a similar meridionally oriented anomaly pattern only in one of the seven simulations presented here (E0, E010, E025). Otherwise, the SLP gradient shows a distinct southwest to northeast orientation, thus dominating the ensemble mean shown in Fig. 13. This means that pronounced anomalies in on-shelf nutrient fluxes are mostly induced by anomalies in the meridional wind component over the NE Atlantic (NAO/SCAN) but can also be triggered by anomalies in the wind speed of the predominant westerlies (NAO).

The impact of GIS meltwater discharge leading to both higher mean pre-bloom nutrient concentrations and annual net primary production in the northern North Sea is significant with respect to the ensemble spreads. The ranges between the ensemble minima and maxima do not overlap among the simulations with and without GIS meltwater discharge (Fig. 15). The increase in the variability, however, is not significant as the ensemble spreads clearly overlap for both variables' CE6 nutrient concentration and primary production. This means that while we have gained understanding about the combined effect of increasing radiative forcing and GIS meltwater discharge on physical processes governing the variability of ocean–shelf exchange, whether or not the related changes would have a significant impact on the shelf conditions is strongly influenced by natural variability.

These effects on the ocean–shelf nutrient exchange have been investigated by ensemble simulations with 0.1 and 0.25 Sv meltwater discharge along the coast of Greenland. Future changes in the Greenland surface mass balance and peripheral glaciers projected by ice sheet models show a similarly substantial spread, ranging from no change at all to twice the ensemble mean (Church et al., 2013; Agarwal et al., 2015; Vizcaino et al., 2015). The higher discharge experiment (E025) is additionally motivated by indications that ice sheet models might underestimate future GIS melting rates (Burkett et al., 2014). Moreover, our downscaling model system as well as the parent global model use a fixed GIS grounding line and topography and are therefore not able to account for all positive feedbacks between runoff-induced changes in the ocean and atmosphere and the melting of the GIS (Driesschaert et al., 2007; Castro de la Guardia et al., 2015). They also do not account for additional meltwater discharge due to icebergs, which brings more freshwater to the Labrador Sea and the interior of the SPG (Marson et al., 2018). Besides, in global circulation models (GCMs), the simulated AMOC weakening varies between 10 % and > 60 % after 100 years for a typical freshwater hosing experiment of 0.1 Sv to the northern North Atlantic (e.g., Stouffer et al., 2006; Swingedouw et al., 2015; Yu et al., 2016, see also Sect. 2.3). Reasons for the uncertainty in simulated AMOC

responses have been linked to the grid resolution of the ocean model (Swingedouw et al., 2013; Weijer et al., 2012; Böning et al., 2016) and the implemented mixing schemes and convection parameterizations (Nilsson et al., 2003; Yu et al., 2008; Marzeion and Levermann, 2009), affecting the simulated export of the added freshwater from the SPG region to adjacent ocean basins. These uncertainties related to GIS meltwater discharge have to be assessed in addition to the uncertainties in the atmospherically driven AMOC weakening and stratification strengthening deduced from multi-model ensembles. Under scenario RCP8.5, the AMOC weakening by the year 2100 ranges between 12 % and 54 % of the individual CMIP5 model's historical mean (Weaver et al., 2012; Cheng et al., 2013), and the upper-ocean stratification increases by 6 %–30 % (Fu et al., 2016) with a similar uncertainty in MLD shoaling (Bopp et al., 2001; Capotondi et al., 2012; Yool et al., 2015; Alexander et al., 2018).

Though in the present study we have not utilized a GCM of typical CMIP or IPCC configurations, we expect comparable uncertainties in the projected AMOC weakening, stratification strengthening and MLD decrease in our downscaling simulations due to the influences of the involved particular model components and the particular forcing global model. These uncertainties are mainly associated with the magnitudes of the simulated changes, since the directions of the changes are consistent within the cited multi-model ensembles. Low uncertainty is therefore associated with the qualitative impacts of GIS meltwater discharge and the involved driving mechanisms, i.e., the general intensification of the regime shift due to stronger stratification, shallower MLD and higher subpycnocline nutrient concentrations. Nevertheless, relative to the change signals, the biases and uncertainties in our model results are partly of opposite directions. The simulated North Atlantic circulation is more sensitive to the atmospheric forcing but less sensitive to additional freshwater input, compared to other climate system models (Sect. 2.3). Present-day wintertime MLD seems to be overestimated but there is no consistent relationship between MLD biases and other variables such as water density or salinity (Heuzé, 2017). It is therefore hard to speculate about different regime shift characteristics as simulated by other climate system models, calling for the need to conduct further downscaling experiments and intermodel comparisons.

In the parent global model (MPI-ESM-LR), the persistent dominance of Atlantic upper-ocean conditions on shaping shelf conditions shown by M19 is also seen in the respective freshwater hosing experiments carried out in preparation of the downscaling simulations presented here (Fig. 16). As the future stratification strengthens and the MLD decreases when GIS meltwater is added, surface nutrient concentrations in the NE Atlantic decrease to lower and lower levels, driving a stronger nutrient decline on the NWES as well. The meltwater impact though is generally very weak, as has also been found, e.g., by Zickfeld et al. (2008) and Kuhlbrodt et al. (2009). The regime shift in the on-shelf nutrient trans-

port cannot be captured by the global model because of its coarse grid resolution (50–70 km at the shelf break) and the neglect of intertidal currents.

## 5 Conclusions

We have made recent RCP8.5 downscaling simulations for the physical and biogeochemical state of the NWES (Mathis et al., 2018, 2019) more consistent by taking into account the effects of GIS meltwater discharge. The considered GIS melting rates cover a wide range projected by ice sheet models (Burkett et al., 2014). Furthermore, we gained knowledge about the influence of the parent global model's internal variability on the characteristics of the proposed regime shift in Atlantic nutrient supply by utilizing an ensemble of three realizations for experiments with 0 and 0.1 Sv GIS melting rates.

We found that for the high-emission RCP8.5 scenario, GIS meltwater discharge influences the nutrient supply from the NE Atlantic to the NWES. It potentially intensifies the regime shift near the end of the 21st century in terms of spatial distribution and temporal variability and leads to an earlier onset, depending on GIS melting rates. Nevertheless, most striking features associated with the regime shift are induced by changes in upper-ocean conditions of the NE Atlantic tied to a critical reduction of the wintertime MLD, a slowdown of the AMOC and a widening of the SPG. These are pronounced changes already in the RCP8.5 simulations without additional GIS meltwater discharge, primarily caused by a thermally induced buoyancy reduction (Liu et al., 2017). The impact of GIS meltwater discharge on the regime shift characteristics is therefore not as substantial as the general difference between the present-day deep-ML and future shallow-ML regimes. The simulated intensity and timing of the regime shift, however, are subject to uncertainties in the projected GIS melting rate, AMOC weakening and NE Atlantic stratification strengthening and MLD shoaling. In particular, it has been demonstrated that under sufficiently shallow wintertime MLDs, changes in the upper-ocean conditions of the NE Atlantic are no longer transmitted to the NWES with an intimate correspondence. Rather, the conditions of on-shelf transport across the shelf break become heavily modified by subpycnocline water masses and hence controlled by the interplay between the deeper ocean, the ocean mixed layer and the atmospheric forcing.

In open shelf areas, the intensification of the regime shift due to GIS meltwater discharge leads to higher mean pre-bloom concentrations and fuels higher productivity during the following plankton growing season (relative to experiments without GIS meltwater discharge). The impact on the variability in nutrient concentrations and productivity, however, is low. The main reasons are a strong influence of less variable Atlantic subpycnocline water masses on the net on-shelf nutrient transport and the influence of local physical

conditions of the upper water column on the shelf (SST and depth of seasonal stratification) on the plankton spring bloom and summer growth (M19). The ensemble approach allowed us to render the impact on the variability insignificant with respect to the ensemble spread. The higher mean pre-bloom concentrations and annual primary production, by contrast, were revealed to be robust signals.

The increasing influence of Atlantic subpycnocline water masses on the NWES physical and biogeochemical state affects not only the supply of oceanic nutrients to the shelf. Any conservative constituent of the on-shelf transport potentially has an impact on the NWES marine ecosystem, such as salinity, temperature, oxygen and alkalinity. Further simulations with regional ecosystem models driven by boundary conditions from high-resolution future projections as presented here are therefore highly appreciated to more comprehensively assess the impact of the shallow-ML regime on the NWES biological environment and the efficacy of the shelf carbon pump.

## Appendix A

**Table A1.** Mean and standard deviation of pre-bloom (March) phosphate concentration and annual net primary production in the northern North Sea (see Fig. 4) during historical conditions as well as the shallow-ML regime for experiments without (E0) and with 0.1 (E010), and 0.25 Sv GIS meltwater discharge. For E0 and E010, values refer to ensemble means with ensemble spreads given in brackets.

| Experiment | Mean March $PO_4$ (mmol P m$^{-3}$) | Mean annual NPP (mol C m$^{-2}$ yr$^{-1}$) | $\sigma$ March $PO_4$ (mmol P m$^{-3}$) | $\sigma$ Annual NPP (mol C m$^{-2}$ yr$^{-1}$) |
|---|---|---|---|---|
| E0 (1971–2000) | 0.736 (0.729–0.743) | 10.49 (10.35–10.65) | 0.0194 (0.0175–0.0213) | 0.382 (0.340–0.406) |
| E0 (2101–2150) | 0.498 (0.492–0.503) | 7.15 (7.12–7.21) | 0.0271 (0.0260–0.0290) | 0.430 (0.409–0.451) |
| E010 (2101–2150) | 0.516 (0.510–0.521) | 7.43 (7.33–7.49) | 0.0335 (0.0269–0.0385) | 0.482 (0.433–0.508) |
| E025 (2101–2150) | 0.530 | 7.54 | 0.0281 | 0.444 |

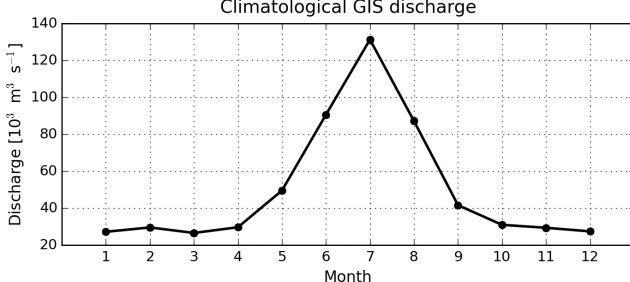

**Figure A1.** Seasonal cycle (monthly means) of GIS discharge following the observational climatology by Martin et al. (2019). The spatial distribution of the annual mean is shown in Fig. 2.

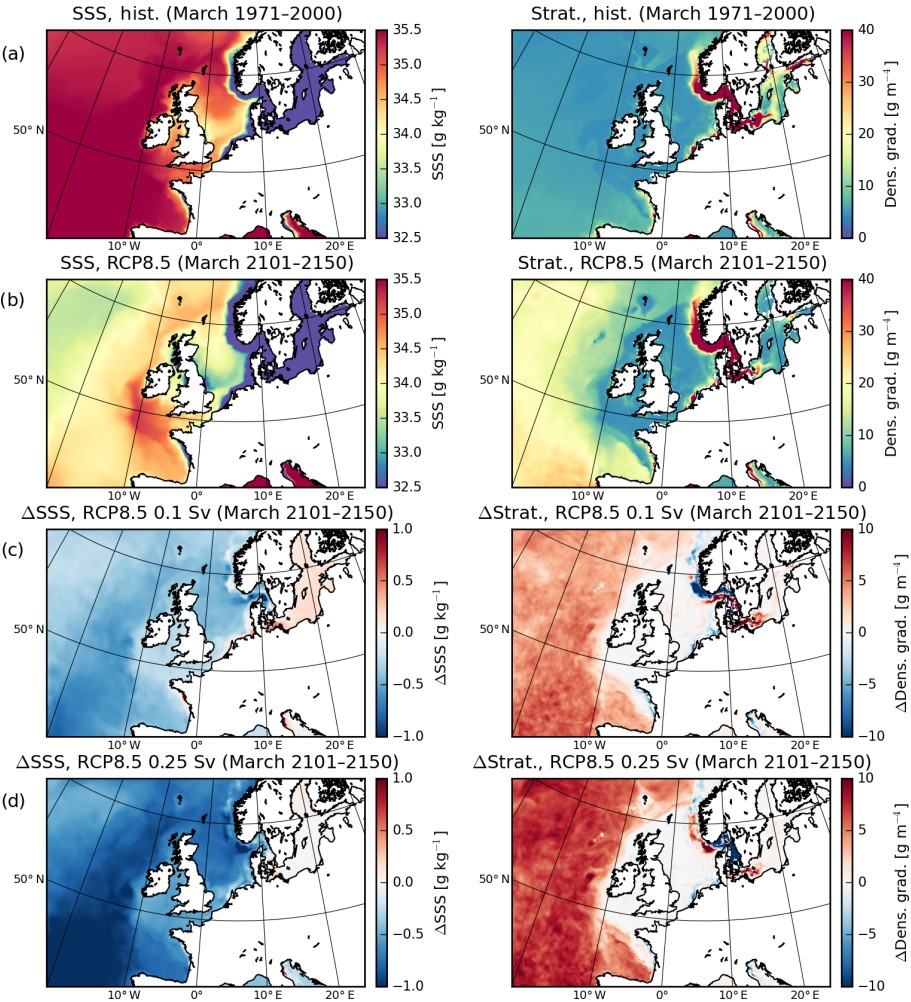

**Figure A2.** Sea surface salinity and strength of stratification at the end of winter (March) for historical (1971–2000, **a**) and RCP8.5 (2101–2150, E0, **b**) conditions. Changes in RCP8.5 conditions (2101–2150) for experiments with 0.1 Sv (E010, **c**) and 0.25 Sv (E025, **d**) GIS meltwater discharge relative to the experiment without GIS meltwater discharge (E0, **b**).

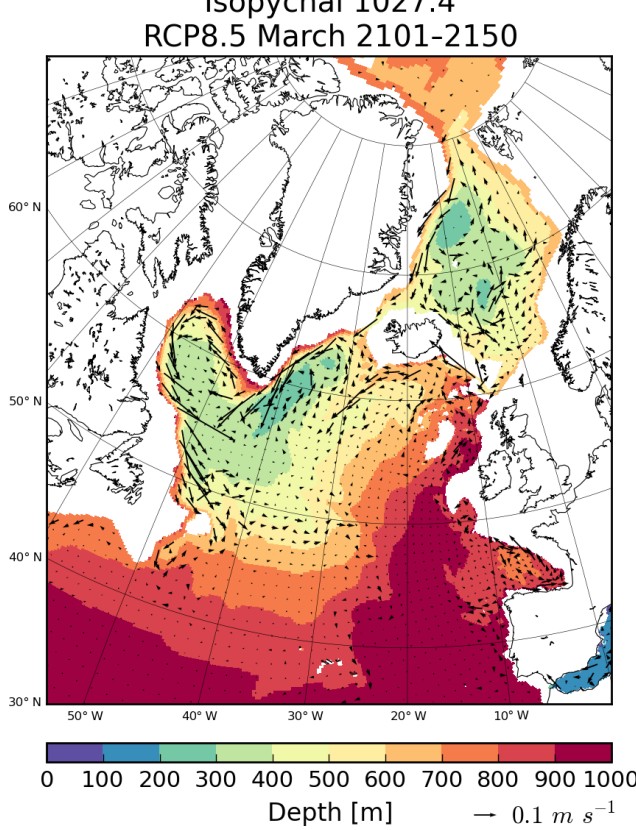

**Figure A3.** Depth of the 1027.4 isopycnal in March under RCP8.5 conditions (2101–2150, E0), with the density of 1027.4 kg m$^{-3}$ being representative of intermediate depths (500–1000 m) in the NE Atlantic sector (45–62° N, 15–28° W). Vectors illustrate the corresponding flow field at the isopycnal.

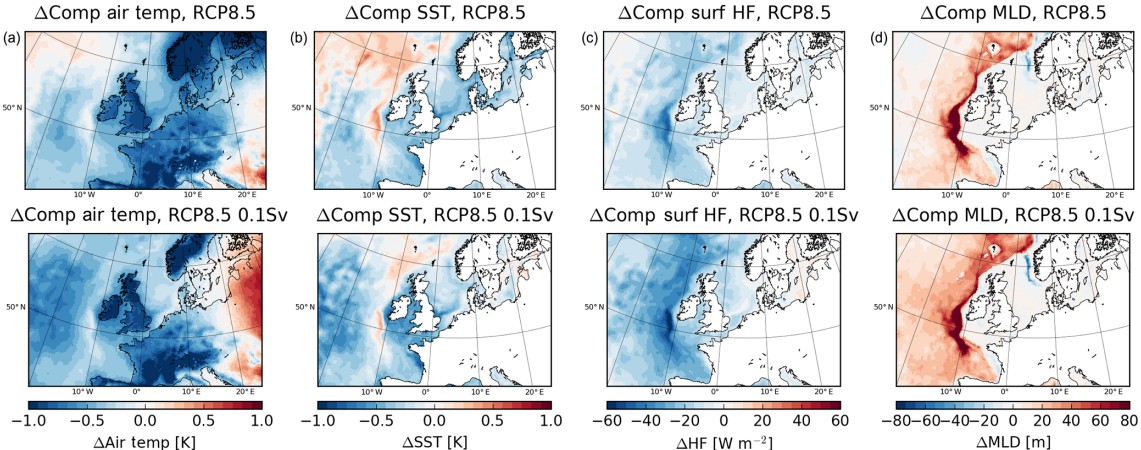

**Figure A4.** Differences between positive and negative composites of 2 m air temperature **(a)**, sea surface temperature **(b)**, downward surface heat flux **(c)** and MLD **(d)** during the shallow-ML regime (2101–2150) for experiments without (E0, upper row) and with (E010, lower row) GIS meltwater discharge. Composites are calculated the same way as for Fig. 13.

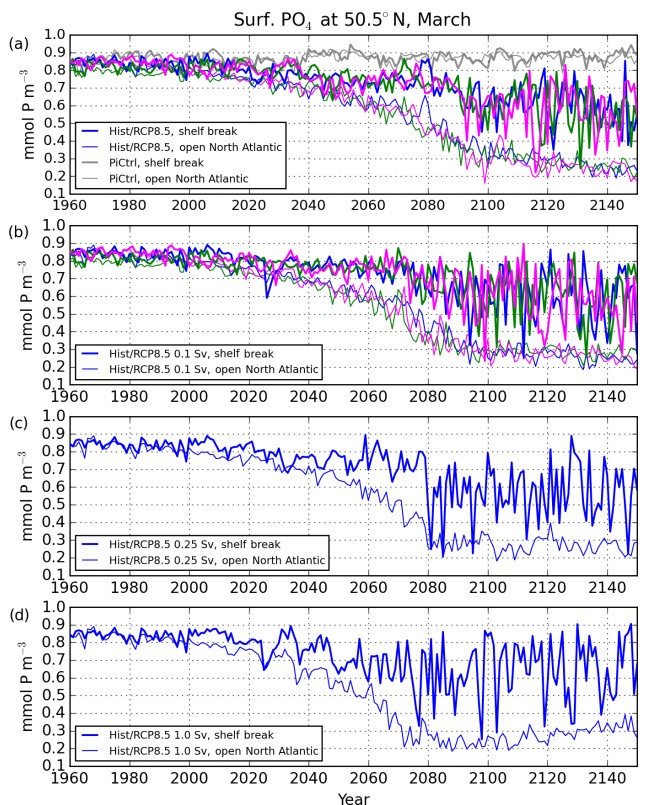

**Figure A5.** Pre-bloom (March) surface phosphate concentration at the Celtic shelf break (50.5° N, 10–12° W, thick lines) and in the open NE Atlantic (50.5° N, 15–20° W, thin lines) for the control run C0 (gray) and individual realizations (blue, green, magenta) of experiments E0 **(a)**, E010 **(b)**, E025 **(c)** and E100 **(d)**.

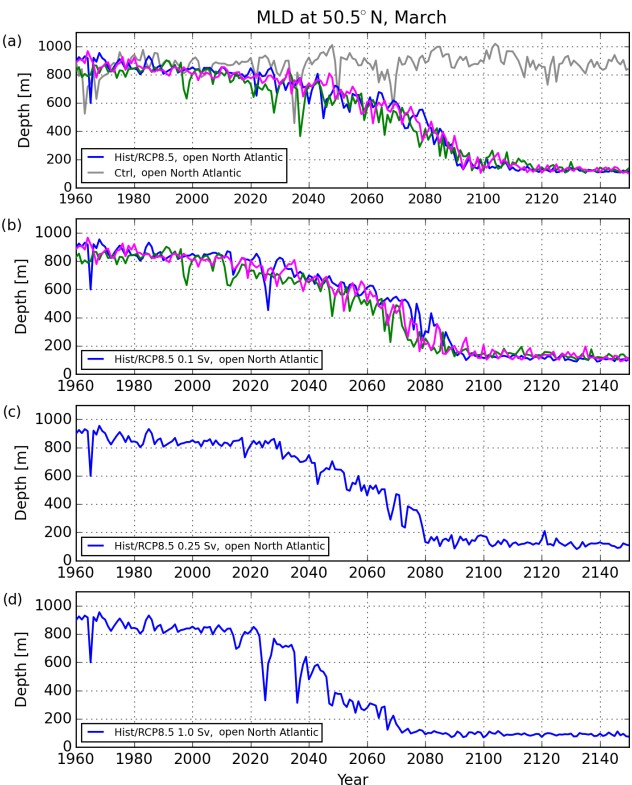

**Figure A6.** March MLD in the NE Atlantic (50.5° N, 15–20° W, thick lines) for the control run C0 (gray) and individual realizations (blue, green, magenta) of experiments E0 **(a)**, E010 **(b)**, E025 **(c)** and E100 **(d)**.

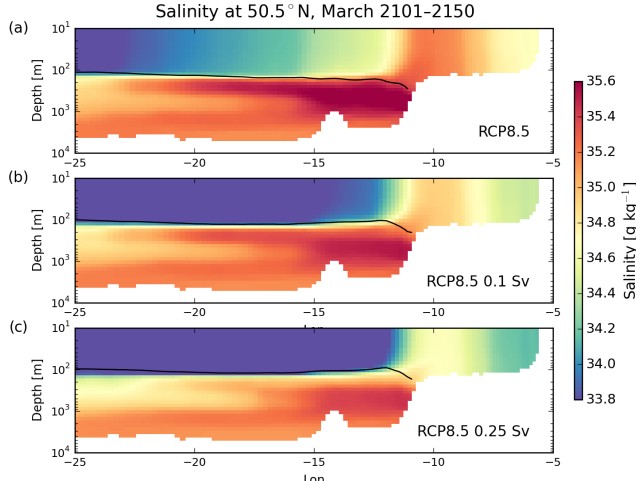

**Figure A7.** Vertical salinity distribution at 50.5° N during the shallow-ML regime (2101–2150) for experiments E0 **(a)**, E010 **(b)** and E025 **(c)**. Black contour line represents mean MLD.

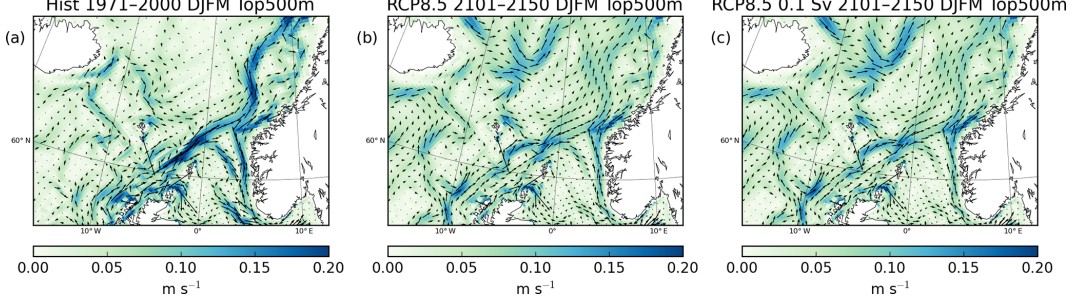

**Figure A8.** Mean wintertime (DJFM) upper-ocean circulation (top 500 m) in the Norwegian Sea during the historical period (1971–2000) **(a)** and the shallow-ML regime (2101–2150) according to experiments E0 **(b)** and E010 **(c)**.

*Code and data availability.* Primary data and scripts used in the analysis that may be useful in reproducing the work are archived by the Max-Planck-Institute for Meteorology and can be obtained by contacting publications@mpimet.mpg.de.

*Author contributions.* MM and UM jointly conceived and designed the model experiments. Simulations and analyses were carried out by MM. Results and conclusions were developed and discussed by both authors. The manuscript was written by MM with contributions from UM.

*Competing interests.* The authors declare that they have no conflict of interest.

*Acknowledgements.* This work was funded by the BMBF-funded joint research projects RACE – Regional Atlantic Circulation and Global Change and RACE – Synthesis. Computational resources were made available by the German Climate Computing Center (DKRZ) through support from the German Federal Ministry of Education and Research (BMBF). The used GIS meltwater climatology was provided by Torge Martin from the Helmholtz Centre for Ocean Research Kiel (GEOMAR).

*Financial support.* This research has been supported by the German Federal Ministry of Education and Research (BMBF) (grant nos. 03F0729D and 03F0824D).

The article processing charges for this open-access publication were covered by a Research Centre of the Helmholtz Association.

*Review statement.* This paper was edited by Markus Meier and reviewed by two anonymous referees.

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

**Remarks from the language copy-editor**

**Remarks from the typesetter**

**TS1**      Due to the requested changes, we have to forward your requests to the handling editor for approval. To explain the corrections needed to the editor, please send me the reason why these corrections are necessary. Even if you say that these changes do not affect any results, the decision of your editor after discussion read "publish as is". Please note that the status of your paper will be changed to "Post-review adjustments" until the editor has made their decision. We will keep you informed via email.