# Peer review of "The impact of melt water discharge from the Greenland ice sheet on the Atlantic nutrient supply to the Northwest European Shelf"

_Ocean Science, 2019_

## Referee Comment (RC1) · Anonymous Referee #1 · 22 Oct 2019

In this study the authors investigated the impact GIS meltwater has on future nutrient supply on the NW European Shelf Seas (NWS). Under RCP8.5, the upper North Atlantic is projected to freshen and warm, with a shoaling MLD, reducing surface nutrient levels. Increased GIS discharge is projected to shoal the MLD further. The NWS shelf edge processes allow some mixing of sub pycnocline water to the surface, which can decouple the NWS nutrient regime from the adjacent surface NE Atlantic. The focus of this paper is to investigate how the varying changes and mechanisms will affect the NWS. I thought the paper was novel, through and would recommend it for publication

with minor corrections.

Review criteria

1. Does the paper address relevant scientific questions within the scope of OS?

Yes.

2. Does the paper present novel concepts, ideas, tools, or data?

As far as I know, yes.

3. Are substantial conclusions reached?

Yes.

4. Are the scientific methods and assumptions valid and clearly outlined?

Yes.

5. Are the results sufficient to support the interpretations and conclusions?

Yes.

6. Is the description of experiments and calculations sufficiently complete and precise to allow their reproduction by fellow scientists (traceability of results)?

Yes, although I have made a few suggestions (location of hosing, figure of model domain etc.

7. Do the authors give proper credit to related work and clearly indicate their own new/original contribution?

Yes

8. Does the title clearly reflect the contents of the paper?

Yes

9. Does the abstract provide a concise and complete summary?

Yes

10. Is the overall presentation well-structured and clear?

Yes

11. Is the language fluent and precise?

Yes

12. Are mathematical formulae, symbols, abbreviations, and units correctly defined and used?

Yes

13. Should any parts of the paper (text, formulae, figures, tables) be clarified, reduced, combined, or eliminated?

No

14. Are the number and quality of references appropriate?

Yes

15. Is the amount and quality of supplementary material appropriate?

Yes

Comments

The authors use a GCM with varying levels of GIS discharge to provide surface boundary conditions for their global ocean model with regional focusing to simulate the response to the GIS discharge. They run a control simulation, a 3-member ensemble under rcp8.5 (from 1920 – 2150) with no hosing, a 3-member ensemble with (a linear increase to) 0.1Sv hosing, and two additional runs with 0.25 and 1Sv. Their methodology is sound, and it good to see they use of (small) ensembles rather than individual

simulations.

There are lots of papers on Atlantic Hosing studies, and I felt this should be discussed more in the introduction. I think this paper needed to have a section comparing your model simulations to these other studies. I am not aware of anyone looking at the impact of hosing on the NWS nutrient supply, so this study is novel, but it would be good to see, for example, that the MLD/AMOC/SSS etc response, is consistent with other studies etc. Through the text you occasionally make statements about this, but I think you should be clear about it at the start. You want the reader to know that your model results are applicable beyond just your model, but to the real world. To do this, you want to show them that your model behaviour is typical (or atypical) – perhaps put into context of Swingedouw et al. 2013 (https://link.springer.com/article/10.1007%2Fs00382-012-1479-9). A couple of other hosing papers that mention changes in marine primary productivity which may be of relevance:

https://link.springer.com/article/10.1007%2Fs10584-009-9561-y

https://agupubs.onlinelibrary.wiley.com/doi/full/10.1029/2007GB003118

I think you should have a figure showing the model domain, outlining model cross-sections – even if this is in the additional material. I don't think it's enough to refereeing to M19 for such a fundamental point. I'd even be happy with it in the appendices.

Clarify where the GIS meltwater is added geographically. Equally around the coast of Greenland?

Perhaps refer to MLD anomalies as deeper, deepening/shoaling etc rather than positive/negative?

Figure 10 also looks like the East Atlantic/West Russian Pattern e.g. Roberts et al. 2016 (http://dx.doi.org/10.1175/JCLI-D-15-0886.1).

Line 286 – ". . . higher SSS on the NWES than in the NE Atlantic." Is this referring to figure A1.b? if so, refer to it. Higher SSS on the NWES or just in the Celtic Sea?

Line 296 near the Celtic shelf break - where? Add to the new model domain figure? Give e.g. lat/lons?

Line 361 – define inner and outer shelf? On new figure??

Lines 428-431 – Either add a figure showing the ensemble overlapping etc,. or point to where this is shown in the figures. I don't think a table is sufficient.

Line 461 – you haven't mentioned ansatz before.

Captions in the Appendices need some work. Figure A2, define vectors. A3, describe how composites are made, add a, b c etc to the subplots. I assume they were the same as figure 10 – if so, say so. Figure A5 & 4, a & b, maybe clarify that the different colours are different ensemble members. A6, what is the contour.

The English in the manuscript is good, although some sentences can be unusual, and could perhaps do with being edited for clarity. A couple of examples are:

"The related decrease in the density strengthens the stratification and reduces the MLD in addition to the climate change signal from the atmosphere." Would be clearer as "In addition to the climate change signal from the atmosphere, the related decrease in the density strengthens the stratification and reduces the MLD."

Lines 292-294 are unclear. . .

I didn't really focus on typos, but there are a few in the paper, for example: line 49 collapse; line 49 lose; caption for figure 10: pressure. . . etc.

---

## Short Comment (SC1) · 24 Oct 2019

Thank you very much for your constructive review. We will thoroughly incorporate your comments and suggestions in the next manuscript version.

Moritz Mathis

---

## Referee Comment (RC2) · Anonymous Referee #2 · 26 Oct 2019

Review of "The impact of melt water discharge from the Greenland ice sheet on the Atlantic nutrient supply to the Northwest European Shelf" by Moritz Mathis and Uwe Mikolajewicz

Mathis and Mikolajewicz investigate the sensitivity of freshwater discharge from the Greenland Ice Sheet on conditions at the Northwest European Shelf in future model scenarios. They find that increased meltwater discharge results in larger variability at the shelf-break. Subpycnocline nutrient concentration increase and results in increased nutrient fluxes and variability at the shelf break. They find that a regime shift

occurs 1-2 decades earlier depending on the discharge rate.

I find the sensitivity experiments very interesting and the results can contribute to our understanding of the impact of climate change in the northern North Atlantic. However, some aspects of the design of the experiments need to be clarified, including the sources of freshwater discharge, and also I find that some of the interpretations of the results needs to be clarified or modified, as I describe below. Finally, I have some minor comments. When these issues have been clarified I can recommend publication in Ocean Science.

Comments

It is not clear where the increased freshwater discharge (FWD) in the experiments takes place. A reference is made to an unpublished manuscript (Martin et al., 2019) and it is described as following the observational climatology. However, relatively few studies have been made on this issue so more information about the locations of the increased discharge and the actual present day values are needed to fully understand the implications of the sensitivity study. It would be interesting to know how the discharge field scales in comparison with observations, for example related to the studies of Bamber et al., (2017) and Mouginot et al. (2019).

l. 150: The sensitivity study is designed as a linear increase of FWD where the final 0.1 Sv is obtained from an ice sheet model. It is not clear whether this simple linear transient increase is just a simple (ad hoc) model for the changing rate or if it is based on numerical experiments?

l. 161: As far as I know, a value of 1Sv is far above any present estimate of future runoff from GIS ($\sim$20 times the present day value). Has it any relations to estimates of future runoff rates?

l. 295: Time series in Fig. A5 should illustrate the earlier onset of the shallow ML-regime for increasing GIS melting rates. I can not see this. There is hardly any difference, as far as I can see, between the HIST and HIST/0.1Sv. Even the 0.25Sv (only a single realization) is quite similar to the HIST. So either this conclusion is reached based on the 0.25 and 1.0 Sv single-realization experiments or it has to be described more clearly where the difference occur. If the conclusion is based on the two large-discharge rate experiments it should be pointed out that these experiments (both single realizations) imply discharge rates between 5-20 times present day values, and also application of these high rates should be justified further, cf. my comment above.

l. 328-334: The decrease in inflow to the North Sea is in qualitative accordance with the study of Holt et al. (2018). This is a very interesting results. However, it is not clear whether the mechanism for the reduced inflow is the same in the two models. Did the authors calculate the change in stratification and the deformations radius and relate it to the curvature of their coarser bathymetry? If not, I would suggest to include it or, otherwise, it should be clarified that this was not analysed.

l. 333: It is stated that the results are similar to Holt et al. (2018). This may by so in a qualitative sense but it is not clear whether the mechanisms are the same, cf. my comment above. Also there is only a qualitative similarity in the sense that the inflow decrease.

Minor comments:

l. 135: The reason that CMIP5 could not be used because of they were made on another super-computer and hence inconsistent is not clear. What was the relevant problem with the super-computer?

l. 186: change -> changed

l. 232: ..the meridional "density gradient" -> density difference (the units are not gradients).

Table 2: the meridional "density gradient" -> density difference (the units are not gradients).

Table 2: ".. at 500-1000m ..." Is it averaged between 500-1000m?

l. 273: explain "..MLD in the NE Atlantic is lower ...". Do you mean more shallow?

l. 277 - 280: This sentence need to be clarified. It seems to imply a relation between SLP and MLD standard deviations (?) and this need to explained.

l. 315: detailled -> detailed

l. 366-370: The argument that meltwater or iceberg-transported substances can make a significant difference to subpycnocline nutrient-concentrations in the northern North Atlantic is not supported by the studies referred. This needs to be clarified or modified.

l. 384: The contribution to the "nutrient flux" is described. However, there are no calculations of the fluxes. (Do you mean a contribution to PP?)

Table 4: the definition of the area ("the northern North Sea") is not specified.

Fig. A5: in a) and b) only the blue color is described.

References

Bamber, J. L., Tedstone, A. J., King, M. D., Howat, I. M., Enderlin, E. M., van den Broeke, M. R., & Noel, B. (2018). Land ice freshwater budget of the Arctic and North Atlantic Oceans: 1. Data, methods, and results. Journal of Geophysical Research: Oceans, 123. https://doi.org/10.1002/2017JC013605 Mouginot, J., Rignot, E., Bjørk, A. A., Van den Broeke, M., Millan, R., Morlighem, M., et al. (2019). Forty six years of Greenland Ice Sheet mass balance from 1972 to 2018, Proceedings of the National Academy of Sciences, 116, 9239–9244. https://doi.org/10.1073/pnas.1904242116

---

## Short Comment (SC2) · 30 Oct 2019

Thank you very much for your thoughtful comments. We will carefully take them into account in the revision of the manuscript.

Moritz Mathis
* * *

---

## Author Comment (AC1) · 27 Nov 2019

Authors responses to

Anonymous Referee 1

In this study the authors investigated the impact GIS meltwater has on future nutrient supply on the NW European Shelf Seas (NWS). Under RCP8.5, the upper North At-

lantic is projected to freshen and warm, with a shoaling MLD, reducing surface nutrient levels. Increased GIS discharge is projected to shoal the MLD further. The NWS shelf edge processes allow some mixing of sub pycnocline water to the surface, which can decouple the NWS nutrient regime from the adjacent surface NE Atlantic. The focus of this paper is to investigate how the varying changes and mechanisms will affect the NWS. I thought the paper was novel, through and would recommend it for publication with minor corrections.

Comments

The authors use a GCM with varying levels of GIS discharge to provide surface boundary conditions for their global ocean model with regional focusing to simulate the response to the GIS discharge. They run a control simulation, a 3-member ensemble under rcp8.5 (from 1920 – 2150) with no hosing, a 3-member ensemble with (a linear increase to) 0.1Sv hosing, and two additional runs with 0.25 and 1Sv. Their methodology is sound, and it good to see they use of (small) ensembles rather than individual simulations.

There are lots of papers on Atlantic Hosing studies, and I felt this should be discussed more in the introduction. I think this paper needed to have a section comparing your model simulations to these other studies. I am not aware of anyone looking at the impact of hosing on the NWS nutrient supply, so this study is novel, but it would be good to see, for example, that the MLD/AMOC/SSS etc response, is consistent with other studies etc. Through the text you occasionally make statements about this, but I think you should be clear about it at the start. You want the reader to know that your model results are applicable beyond just your model, but to the real world. To do this, you want to show them that your model behaviour is typical (or atypical) – perhaps put into context of Swingedouw et al. 2013 (https://link.springer.com/article/10.1007

https://link.springer.com/article/10.1007https://agupubs.onlinelibrary.wiley.com/doi/full/10.1029/2007GB003061)

R: We added a dedicated model evaluation section (section 2.3), putting our model

results into context of observations and other model studies. Specifically, we consider present-day conditions and projected changes in MLD, strength of AMOC, SSS, SST, nutrient concentrations and primary production. In the introduction, we mainly reflect on expected meltwater impacts on the MLD, which are essential to understand the motivation of the study.

I think you should have a figure showing the model domain, outlining model crosssections – even if this is in the additional material. I don't think it's enough to refereeing to M19 for such a fundamental point. I'd even be happy with it in the appendices.

R: We added a respective figure as Fig. 4.

Clarify where the GIS meltwater is added geographically. Equally around the coast of Greenland?

R: We added another figure showing the spatial distribution of the freshwater discharge around the coast of Greenland (Fig. 2).

Perhaps refer to MLD anomalies as deeper, deepening/shoaling etc rather than positive/negative?

R: The term "anomaly" is partiularly useful here because it refers to a deviation from the mean state. Using deeper/shallower or deepening/shallowing instead would be prone to confusion with future changes or differences between experiments and realizations. Besides, "positive MLD anomaly" is only used once. To facilitate understanding, we added "(deepening)" in brackets.

L61: "In particular, a positive MLD anomaly (deepening) leads to an erosion of warm and saline subpycnocline water masses and initiates a positive feedback on the surface heat flux, the upper ocean buoyancy and the MLD."

Figure 10 also looks like the East Atlantic/West Russian Pattern e.g. Roberts et al. 2016 (http://dx.doi.org/10.1175/JCLI-D-15-0886.1).

R: Indeed, the western part of the EA-WR pattern is very similar. The other two characteristic pressure centers over West Siberia and China, however, are not shown in the larger structure of the SLP anomaly simulated by MPI-ESM. We added the following:

L479: "It also resembles the European/North Atlantic sector of the East Atlantic-West Russian pattern, a higher-order atmospheric mode of the northern hemisphere which occasionally mixes with the NAO (Barnston and Livezey, 1987; Fagherazzi et al., 2005). The other two characteristic pressure centers over West Siberia and China, however, are not shown in the larger SLP anomaly structure corresponding to Fig. 13 (simulated by MPI-ESM; not shown)."

Line 286 – "... higher SSS on the NWES than in the NE Atlantic." Is this referring to figure A1.b? if so, refer to it. Higher SSS on the NWES or just in the Celtic Sea?

R: True, only in the Celtic Sea SSS becomes higher than in the NE Atlantic. We changed this sentence and added a reference to Fig. A2 (former A1).

L358: "Moreover, in all experiments the mixing of saline subpycnocline water masses to the upper ocean near the shelf break establishes an ocean-shelf salinity front with higher SSS in the Celtic Sea than in the NE Atlantic (Fig. A2)."

Line 296 near the Celtic shelf break - where? Add to the new model domain figure? Give e.g. lat/lons?

R: We added a new figure showing the model domain and respective cross sections (Fig. 4) and refer to it in the relevant figure captions.

Line 361 – define inner and outer shelf? On new figure??

R: This is meant as a qualitative distinction related to the influences of Atlantic inflow (outer shelf) and river runoff (inner shelf). We added this information in brackets.

L449: "In addition to the decrease in the nutrient transport across the shelf break, this further intensifies the mean nutrient gradient between the inner (dominated by river

loads) and outer (dominated by Atlantic inflow) shelf areas as well as its interannual variability."

Lines 428-431 – Either add a figure showing the ensemble overlapping etc,. or point to where this is shown in the figures. I don't think a table is sufficient.

R: We have transferred the table into a new figure (Fig. 15) and moved the table to the appendix (Table A1) to also provide exact values.

Line 461 – you haven't mentioned ansatz before.

R: We changed "ansatz" to the synonym "approach" (L560).

Captions in the Appendices need some work. Figure A2, define vectors. A3, describe how composites are made, add a, b c etc to the subplots. I assume they were the same as figure 10 – if so, say so. Figure A5 4, a  b, maybe clarify that the different colours are different ensemble members. A6, what is the contour.

R: All figures have been updated according to the reviewers suggestions.
Fig. A2 became Fig. A3
Fig. A3 -> Fig. A4 (Fig. 10 -> Fig. 13)
Fig. A4 -> Fig. A5
Fig. A5 -> Fig. A6
Fig. A6 -> Fig. A7

The English in the manuscript is good, although some sentences can be unusual, and could perhaps do with being edited for clarity. A couple of examples are:

"The related decrease in the density strengthens the stratifcation and reduces the MLD in addition to the climate change signal from the atmosphere." Would be clearer as "In addition to the climate change signal from the atmosphere, the related decrease in the density strengthens the stratifcation and reduces the MLD."

R: This sentence has been changed according to the reviewer's suggestion. Moreover,

the English of the entire ms and in particular the structure of sentences have been polished by a colleague with particularly profound English skills.

Lines 292-294 are unclear...

R: We modified this paragraph to be more clear:

L366: "The shallow-ML regime is fully established when the ML near the shelf break becomes as shallow as the depth of the shelf edge, i.e. about 150-200 m. As can be expected, the impact of GIS meltwater discharge on the stratification strengthening and ML shoaling (section 3.1) implies that the MLD in the NE Atlantic decreases faster when GIS meltwater is added, passing this critical depth earlier in the 21st century (Fig. A6)."

I didn't really focus on typos, but there are a few in the paper, for example: line 49 collapse; line 49 lose; caption for figure 10: pressure... etc.

R: Typos have been removed in the course of the English polishing.

---

## Author Comment (AC2) · 27 Nov 2019

Authors responses to

Anonymous Referee 2

Mathis and Mikolajewicz investigate the sensitivity of freshwater discharge from the Greenland Ice Sheet on conditions at the Northwest European Shelf in future model

scenarios. They find that increased meltwater discharge results in larger variability at the shelf-break. Subpycnocline nutrient concentration increase and results in increased nutrient fluxes and variability at the shelf break. They find that a regime shift occurs 1-2 decades earlier depending on the discharge rate.

I find the sensitivity experiments very interesting and the results can contribute to our understanding of the impact of climate change in the northern North Atlantic. However, some aspects of the design of the experiments need to be clarified, including the sources of freshwater discharge, and also I find that some of the interpretations of the results needs to be clarified or modified, as I describe below. Finally, I have some minor comments. When these issues have been clarified I can recommend publication in Ocean Science.

Comments

It is not clear where the increased freshwater discharge (FWD) in the experiments takes place. A reference is made to an unpublished manuscript (Martin et al., 2019) and it is described as following the observational climatology. However, relatively few studies have been made on this issue so more information about the locations of the increased discharge and the actual present day values are needed to fully understand the implications of the sensitivity study. It would be interesting to know how the discharge field scales in comparison with observations, for example related to the studies of Bamber et al., (2017) and Mouginot et al. (2019).

R: We added a figure showing the spatial distribution of freshwater discharge along the coast of Greenland (Fig. 2). Furthermore, we added more information about the used climatology and compare it with other observational data.

L133: "The spatial distribution of the runoff (Fig. 2) follows the climatology by Bamber et al. (2012), based on satellite observations and regional climate modeling. The seasonal cycle (Fig. A1) has been derived by Martin et al. (2019). The annual mean GIS freshwater flux according to this study corresponds to 0.05 Sv and is comparable

to the estimate of about 0.04 Sv since the year 2010 by Yang et al. (2016) and Bamber et al. (2018). In our simulations, the prescribed freshwater fluxes enter the surface layer of the ocean model, thus ignoring that many marine-terminating outlet glaciers have a grounding line depth several hundred meters below sea level (An et al., 2017; Morlighem et al., 2017)."

l. 150: The sensitivity study is designed as a linear increase of FWD where the final 0.1 Sv is obtained from an ice sheet model. It is not clear whether this simple linear transient increase is just a simple (ad hoc) model for the changing rate or if it is based on numerical experiments?

R: We added more information about this approach.

L160: "The assumption of a linear increase is an idealized approach to deal with the uncertainty in the construction of a hydrological sensitivity parameter, often defined as a constant freshwater discharge per degree atmospheric warming (e.g. Zickfeld et al., 2008; Kuhlbrodt et al., 2009), and has likewise been applied e.g. in Jungclaus et al. (2006)."

l. 161: As far as I know, a value of 1Sv is far above any present estimate of future runoff from GIS ( 20 times the present day value). Has it any relations to estimates of future runoff rates?

R: Indeed, this high melting rate goes beyond any estimates for the 21st century. Experiment E100 was designed only to better understand the processes that limit the freshwater impact on the regime shift timing. We extended the explanation of this experiment in section 2.2.

L175: "This high discharge rate is purely motivated by process understanding and exceeds any present estimate of GIS runoff during the 21st century. In fact, given a present-day GIS volume of about $2.9 \times 10^{15}$ m3, it would lead to a complete disintegration of the ice sheet in the first half of the 22nd century, depending on the surface

mass balance."

l. 295: Time series in Fig. A5 should illustrate the earlier onset of the shallow MLregime for increasing GIS melting rates. I can not see this. There is hardly any difference, as far as I can see, between the HIST and HIST/0.1Sv. Even the 0.25Sv (only a single realization) is quite similar to the HIST. So either this conclusion is reached based on the 0.25 and 1.0 Sv single-realization experiments or it has to be described more clearly where the difference occur. If the conclusion is based on the two large discharge rate experiments it should be pointed out that these experiments (bothsingle realizations) imply discharge rates between 5-20 times present day values, and also application of these high rates should be justified further, cf. my comment above.

R: We agree with the reviewer that an impact of 10-20 years was somewhat overestimated. We removed this estimate from the conclusions, and in the results (section 3.3) we focus more on the generally weak impact on the regime shift timing. In addition to Fig. A6 (former A5) we refer to the PO4 time series shown in Fig. 12 and 16 to support the interpretation for experiment E010 showing an earlier onset by about 10 years.

L369: "Nevertheless, the changes in the regime shift timing are surprisingly weak, given the melting rates vary considerably between the experiments. For a melting rate of 0.1 Sv (E010) the impact is only about 10 years (see also PO4 time series shown in Fig. 12 and 16)."

l. 328-334: The decrease in inflow to the North Sea is in qualitative accordance with the study of Holt et al. (2018). This is a very interesting results. However, it is not clear whether the mechanism for the reduced inflow is the same in the two models. Did the authors calculate the change in stratification and the deformations radius and relate it to the curvature of their coarser bathymetry? If not, I would suggest to include it or, otherwise, it should be clarified that this was not analysed.

R: We don't think a comparison with the curvature of our model topography is meaningful here. Holt et al. have shown that the geostrophic component of the Norwegian Trech

inflow scales well with the deformation radius, supporting the general explanation that a relaxation of topographic steering leads to a reduction of the inflow. Accordingly, we added simulated changes in the stratification and deformation radius near the entrance of the Norwegian inflow as well as the correlation between the deformation radius and the strength of the inflow. Furthermore, we found an intensification of the Nordic Seas circulation, in particular the southeastward flowing branch of the East Iceland Current, in line with Holt et al. and added a figure showing this (Fig. A8). We thus confirm the increasing influence of fresh Arctic water masses on the Norwegian Trench circulation but also point out the different timing in our simulations as well as the missing reversal of the inflow to a persistent outflow.

L401-422: "The inflow along the western side of the Norwegian Trench is mainly governed by topographic steering as the slope current follows the sharp topographic turn to the right at about 62°N. Our simulations indicate a substantial weakening of this inflow (Fig.14c) and are in line with the findings by Holt et al. (2018). The proposed driving mechanism is an increase in the deformation radius due to the strengthening of the stratification. In addition to the general weakening of the slope current, a smaller fraction of the slope current is then able to follow the topographic turn. In our simulations, the permanent stratification around the entrance of the Norwegian Trench strengthens by about 5 gm-4 in experiment E0 and 7 gm-4 in E010 (Fig.A2), and the baroclinic deformation radius increases by about 4-5 km in E0 (similar to 3-4 km in Holt et al., 2018), and 7 km in E010. The maximum deformation radii at the end of the simulations are about 10 and 12 km, respectively. During the shallow-ML regime (2101-2150), the correlation between the Norwegian Trench inflow and the deformation radius is about -0.35 (detrended time series) for both experiments. After around 2120, the remaining inflow to the Norwegian Trench does not penetrate further south than about 60°N, implying a reduction down to 0 in Fig.14c.

In Holt et al. (2018), the changes in the stratification and deformation radius are amplified by a strengthening of the circulation in the Nordic Seas, in particular the East

[Figure]

Iceland Current, leading to a southward intrusion of cold and fresh Arctic water masses into the northern inflow region of the North Sea. Similar in our simulations, the Nordic Seas circulation strengthens and a larger fraction of the East Iceland Current passes the Faroe Islands southeastward, joining the slope current to the north of the Shetland Islands (Fig.A8). Northeast of the Faroes, the East Iceland Current intensifies by about 30% (in E0 and E010). We thus confirm the enhanced influence of Arctic water masses on the Norwegian Trench inflow projected by Holt et al. (2018). A sudden shutdown due to positive feedback with the accumulation of coastal North Sea water of low salinity, however, does not occur in our simulations. The changes in the circulation are rather smooth over several decades (see Fig.14c for the Norwegian Trench inflow, not shown for the East Iceland Current) and simulated to happen later than in Holt et al. (2018) by about 30 years. Moreover, a reversal of the Norwegian Trench inflow to a persistent outflow, as also projected by Tinker et al. (2016), is not indicated but may happen post 2150."

l. 333: It is stated that the results are similar to Holt et al. (2018). This may by so in a qualitative sense but it is not clear whether the mechanisms are the same, cf. my comment above. Also there is only a qualitative similarity in the sense that the inflow decrease.

R: Our additional analysis mentioned in the response to the previous comment has shown that the mechanisms are indeed qualitatively the same. Due to the quantitative differences, however, we interpreted our results as "in qualitative agreement".

L444: "The maintained connection to nutrient-rich Atlantic subpycnocline water masses gains particular importance as the southward turn of the slope current into the Norwegian Trench is found to weaken substantially during the first half of the 22nd century, thus closing the only direct inflow of deeper Atlantic water to the NWES (in qualitative agreement with Holt et al., 2018)."

Minor comments:

l. 135: The reason that CMIP5 could not be used because of they were made on another super-computer and hence inconsistent is not clear. What was the relevant problem with the super-computer?

R: The high sensitivity of complex earth system models to differences in the representation and accuracy of high precision float numbers on the hosting super computer ultimately leads to independent trajectories already after short simulation times (butterfly effect). We added a sentence to make this more clear.

L143: "The original CMIP5 simulations could not be used here because they were run on a former high-performance computer (HPC) and hence are inconsistent with our GIS discharge experiments. A bitwise reproduction of a simulation is not possible on different HPCs, leading to independent trajectories even when started from identical initial conditions."

l. 186: change -> changed

R: "change" is not used as a verb here but as the term "change signal", commonly inferred from the concept of climate change signal.

L256: "The contrasting impact on the nutrient distribution seen in the meltwater discharge experiments (E010 and E025) indicates that also here the change signal in the NE Atlantic is not coherently transferred to the shelf but there are other mechanisms involved influencing the on-shelf nutrient transport."

l. 232: ..the meridional "density gradient" -> density difference (the units are not gradients).

R: Changed (L305).

Table 2: the meridional "density gradient" -> density difference (the units are not gradients).

R: Changed (Table 2).

Table 2: ".. at 500-1000m ..." Is it averaged between 500-1000m?

R: Changed to "averaged over 500-1000 m depth" (Table 2).

l. 273: explain "..MLD in the NE Atlantic is lower ...". Do you mean more shallow?

R: Changed to "more shallow" (L346).

l. 277-280: This sentence need to be clarified. It seems to imply a relation between SLP and MLD standard deviations (?) and this need to explained.

R: The impact of SLP anomalies over the NE Atlantic on the MLD was a main finding in M19 and is summarized here in L334-344. We added a referring sentence to the caption of Table 3.

"The relation between SLP and MLD anomalies is explained in section 3.2"

l. 315: detailled -> detailed

R: Changed (L387).

l. 366-370: The argument that meltwater or iceberg-transported substances can make a significant difference to subpycnocline nutrient-concentrations in the northern North Atlantic is not supported by the studies referred. This needs to be clarified or modified.

R: The cited studies suggest that glacial runoff from the GIS serves as a significant source of bioavailable nutrients to the surrounding coastal ocean, which is likely to increase as GIS melting escalates under climate warming. Parts of the nutrients released to the upper ocean are consumed by phytoplankton and transferred to deeper levels by export production. We added a sentence for clarification.

L456: "Moreover, the subpycnocline nutrient enrichment may be underestimated because our model system does not account for the effect of biologically relevant substances transported into the ocean by meltwater and iceberg calving due to microbial activity and hydrolysis reactions at the interface between land ice and the bedrock

(such as dissolved iron, silicate and nitrogen; Bhatia et al., 2013; Duprat et al., 2016; Wadham et al., 2016; Hatton et al., 2019). Part of this additional nutrient input to the upper ocean would be consumed by primary producers and exported to deeper levels."

l. 384: The contribution to the "nutrient flux" is described. However, there are no calculations of the fluxes. (Do you mean a contribution to PP?)

R: We added calculated nutrient fluxes to section 3.1 and refer to them respectively.

L295: "Net on-shelf $PO_4$ fluxes decrease from 839 mol s-1 (1971-2000) to 311 mol s-1 (2101-2150) in E0, 265 mol s-1 in E010, and 211 mol s-1 in E025. This increasing reduction among the experiments, however, is dominated by the decreasing volume transports (-0.25 Sv in E0, -0.30 in E010, -0.32 in E025) and does not reflect the changes in the nutrient concentrations."

L475: "The contribution to the mean nutrient flux (section 3.1) though is only about 3% for E010."

Table 4: the definition of the area ("the northern North Sea") is not specified.

R: We introduced a new figure (Fig. 4) showing the bathymetry of the study area, the transect across the Celtic shelf break and the specification of the northern North Sea. We refer to this figure in the relevant figures and tables, as e.g. in Table A1 (former Table 4).

Fig. A5: in a) and b) only the blue color is described.

R: Changed (now Fig. A6).

References

Bamber, J. L., Tedstone, A. J., King, M. D., Howat, I. M., Enderlin, E. M., van den Broeke, M. R., Noel, B. (2018). Land ice freshwater budget of the Arctic and North Atlantic Oceans: 1. Data, methods, and results. Journal of Geophysical Research: Oceans, 123. https://doi.org/10.1002/2017JC013605 Mouginot, J., Rignot, E., Bjørk,

[Figure]

A. A., Van den Broeke, M., Millan, R., Morlighem, M., et al. (2019). Forty six years of Greenland Ice Sheet mass balance from 1972 to 2018, Proceedings of the National Academy of Sciences, 116, 9239–9244. https://doi.org/10.1073/pnas.1904242116